# Personalized Federated Learning with Theoretical Guarantees: A Model-Agnostic Meta-Learning Approach

**Alireza Fallah**
EECS Department
Massachusetts Institute of Technology
Cambridge, MA 02139
afallah@mit.edu

**Aryan Mokhtari**
ECE Department
University of Texas at Austin
Austin, TX 78712
mokhtari@austin.utexas.edu

**Asuman Ozdaglar**
EECS Department
Massachusetts Institute of Technology
Cambridge, MA 02139
asuman@mit.edu

## Abstract

In Federated Learning, we aim to train models across multiple computing units (users), while users can only communicate with a common central server, without exchanging their data samples. This mechanism exploits the computational power of all users and allows users to obtain a richer model as their models are trained over a larger set of data points. However, this scheme only develops a *common* output for all the users, and, therefore, it does not adapt the model to each user. This is an important missing feature, especially given the *heterogeneity* of the underlying data distribution for various users. In this paper, we study a personalized variant of the federated learning in which our goal is to find an *initial shared model* that current or new users can easily adapt to their local dataset by performing one or a few steps of gradient descent with respect to their own data. This approach keeps all the benefits of the federated learning architecture, and, by structure, leads to a more personalized model for each user. We show this problem can be studied within the Model-Agnostic Meta-Learning (MAML) framework. Inspired by this connection, we study a *personalized* variant of the well-known Federated Averaging algorithm and evaluate its performance in terms of gradient norm for non-convex loss functions. Further, we characterize how this performance is affected by the closeness of underlying distributions of user data, measured in terms of distribution distances such as Total Variation and 1-Wasserstein metric.

## 1   Introduction

In Federated Learning (FL), we consider a set of $n$ users that are all connected to a central node (server), where each user has access only to its local data [1]. In this setting, the users aim to come up with a model that is trained over all the data points in the network without exchanging their local data with other users or the central node due to privacy issues or communication limitations. More

formally, if we define $f_i : \mathbb{R}^d \to \mathbb{R}$ as the loss corresponding to user $i$, the goal is to solve

$$\min_{w \in \mathbb{R}^d} f(w) := \frac{1}{n} \sum_{i=1}^{n} f_i(w). \tag{1}$$

In particular, consider a supervised learning setting, where $f_i$ represents expected loss over the data distribution of user $i$, i.e.,

$$f_i(w) := \mathbb{E}_{(x,y) \sim p_i} \left[ l_i(w; x, y) \right], \tag{2}$$

where $l_i(w; x, y)$ measures the error of model $w$ in predicting the true label $y \in \mathcal{Y}_i$ given the input $x \in \mathcal{X}_i$, and $p_i$ is the distribution over $\mathcal{X}_i \times \mathcal{Y}_i$. The focus of this paper is on a data *heterogeneous* setting where the probability distribution $p_i$ of users are not identical. To illustrate this formulation, consider the example of training a Natural Language Processing (NLP) model over the devices of a set of users. In this problem, $p_i$ represents the empirical distribution of words and expressions used by user $i$. Hence, $f_i(w)$ can be expressed as $f_i(w) = \sum_{(x,y) \in \mathcal{S}_i} p_i(x, y) l_i(w; x, y)$, where $\mathcal{S}_i$ is the data set corresponding to user $i$ and $p_i(x, y)$ is the probability that user $i$ assigns to a specific word which is proportional to the frequency of using this word by user $i$.

Indeed, each user can solve its local problem defined in (2) without any exchange of information with other users; however, the resulted model may not generalize well to new samples as it has been trained over a small number of samples. If users cooperate and exploit the data available at all users, then their local models could obtain stronger generalization guarantees. A conventional approach for achieving this goal is minimizing the aggregate of local functions defined in (1). However, this scheme only develops a *common* output for all the users, and therefore, it does not adapt the model to each user. In particular, in the *heterogeneous* settings where the underlying data distribution of users are not identical, the resulted global model obtained by minimizing the average loss could perform arbitrarily poorly once applied to the local dataset of each user. In other words, the solution of problem (1) is not *personalized* for each user. To highlight this point, recall the NLP example, where although the distribution over the words and expressions varies from one person to another, the solution to problem (1) provides a shared answer for all users, and, therefore, it is not fully capable of achieving a user-specific model.

In this paper, we overcome this issue by considering a modified formulation of the federated learning problem which incorporates personalization (Section 2). Building on the Model-Agnostic Meta-Learning (MAML) problem formulation introduced in [2], the goal of this new formulation is to find an initial point shared between all users which performs well *after* each user updates it with respect to its own loss function, potentially by performing a few steps of a gradient-based method. This way, while the initial model is derived in a distributed manner over all users, the final model implemented by each user differs from other ones based on her or his own data. We study a Personalized variant of the FedAvg algorithm, called Per-FedAvg, designed for solving the proposed personalized FL problem (Section 3). In particular, we elaborate on its connections with the original FedAvg algorithm [3], and also, discuss a number of considerations that one needs to take into account for implementing Per-FedAvg. We also establish the convergence properties of the proposed Per-FedAvg algorithm for minimizing non-convex loss functions (Section 4). In particular, we characterize the role of data heterogeneity and closeness of data distribution of different users, measured by distribution distances, such as Total Variation (TV) or 1-Wasserstein, on the convergence of Per-FedAvg.

**Related Work.** Recently we have witnessed significant progress in developing novel methods that address different challenges in FL; see [4, 5]. In particular, there have been several works on various aspects of FL, including preserving the privacy of users [6–9] and lowering communication cost [10–13]. Several work develop algorithms for the homogeneous setting, where the data points of all users are sampled from the same probability distribution [14–17]. More related to our paper, there are several works that study statistical heterogeneity of users' data points in FL [18–23], but they do not attempt to find a *personalized solution* for each user.

The centralized version of model-agnostic meta-learning (MAML) problem was first proposed in [2] and followed by a number of papers studying its empirical characteristics [24–29] as well as its convergence properties [30, 31]. In this work, we focus on the convergence of MAML methods for the FL setting that is more challenging as nodes perform multiple local updates before sending their updates to the server, which is not considered in previous theoretical works on meta-learning.

Recently, the idea of personalization in FL and its connections with MAML has gained a lot of attention. In particular, [32] considers a formulation and algorithm similar to our paper, and elaborates

on the empirical success of this framework. Also, recently, there has been a number of other papers that have studied different combinations of MAML-type methods with FL architecture from an empirical point of view [33, 34]. However, our main focus is on developing a theoretical understating regarding this formulation, where we characterize the convergence of the Per-FedAvg, and the role of this algorithm's parameters on its performance. Besides, in our numerical experiment section, we show how the method studied in [32] may not perform well in some cases, and propose another algorithm which addresses this issue. In addition, an independent and concurrent work [35] studies a similar formulation theoretically for the case of strongly convex functions. The results in [35] are completely different from ours, as they study the case that the functions are strongly convex and exact gradients are available, while we study nonconvex functions, and also address gradient stochasticity.

Using meta-learning and multi-task learning to achieve personalization is not limited to MAML framework. In particular, [36] proposes ARUBA, a meta-learning algorithm inspired by online convex optimization, and shows that applying it to FedAvg improves its performance. A similar idea is later used in [37] to design differentially private algorithms with application in FL. Also in [38], the authors use multi-task learning framework and propose a new method, MOCHA, to address the statistical and systems challenges, including data heterogeneity and communication efficiency. Their proposed multi-task learning scheme also leads to a set of solutions that are more user-specific. A detailed survey on the connections of FL and multi-task and meta-learning can be found in [4, 5]. Also, in [39], the authors consider a framework for training a mixture of a single global model and local models, leading to a personalized solution for each user. A similar idea has been studied in [40], where the authors propose an adaptive federated learning algorithm that learns a mixture of local and global models as the personalized model.

## 2 Personalized Federated Learning via Model-Agnostic Meta-Learning

As we stated in Section 1, our goal in this section is to show how the fundamental idea behind the Model-Agnostic Meta-Learning (MAML) framework in [2] can be exploited to design a personalized variant of the FL problem. To do so, let us first briefly recap the MAML formulation. Given a set of tasks drawn from an underlying distribution, in MAML, in contrast to the traditional supervised learning setting, the goal is not finding a model which performs well on all the tasks in expectation. Instead, in MAML, we assume we have a limited computational budget to update our model after a new task arrives, and in this new setting, we look for an *initialization* which performs well *after* it is updated with respect to this new task, possibly by one or a few steps of gradient descent. In particular, if we assume each user takes the initial point and updates it using one step of gradient descent with respect to its own loss function, then problem (1) changes to

$$\min_{w \in \mathbb{R}^d} F(w) := \frac{1}{n} \sum_{i=1}^{n} f_i(w - \alpha \nabla f_i(w)), \tag{3}$$

where $\alpha \geq 0$ is the stepsize. The strength of this formulation is that, not only it allows us to maintain the advantages of FL, but also it captures the difference between users as either existing or new users can take the solution of this new problem as an initial point and slightly update it with respect to their own data. Going back to the NLP example, this means that the users could take this resulting initialization and update it by going over their own data $\mathcal{S}_i$ and performing just one or few steps of gradient descent to obtain a model that works well for their own dataset.

As mentioned earlier, for the considered heterogeneous model of data distribution, solving problem (1) is not the ideal choice as it returns a single model that even after a few steps of local gradient may not quickly adjust to each users local data. On the other hand, by solving (3) we find an initial model (Meta-model) which is trained in a way that after one step of local gradient leads to a good model for each individual user. This formulation can also be extended to the case that users run a few steps of gradient update, but to simplify our notation we focus on the single gradient update case. We would like to mention that the problem formulation in (3) for FL was has been proposed independently in another work [32] and studied numerically. In this work, we focus on the theoretical aspect of this problem and seek a provably convergent method for the case that the functions $f_i$ are nonconvex.

# 3 Personalized FedAvg

In this section, we present the Personalized FedAvg (Per-FedAvg) method to solve (3). This algorithm is inspired by FedAvg, but it is designed to find the optimal solution of (3) instead of (1). In FedAvg, at each round, the server chooses a fraction of users with size $rn$ ($r \in (0, 1]$) and sends its current model to these users. Each selected user $i$ updates the received model based on its own loss function $f_i$ and by running $\tau \geq 1$ steps of stochastic gradient descent. Then, the active users return their updated models to the server. Finally, the server updates the global model by computing the average of the models received from these selected users, and then the next round follows. Per-FedAvg follows the same principles. First, note that function $F$ in (3) can be written as the average of *meta-functions* $F_1, \ldots, F_n$ where the meta-function $F_i$ associated with user $i$ is defined as

$$F_i(w) := f_i(w - \alpha \nabla f_i(w)). \tag{4}$$

To follow a similar scheme as FedAvg for solving problem (3), the first step is to compute the gradient of local functions, which in this case, the gradient $\nabla F_i$, that is given by

$$\nabla F_i(w) = \left(I - \alpha \nabla^2 f_i(w)\right) \nabla f_i(w - \alpha \nabla f_i(w)). \tag{5}$$

Computing the gradient $\nabla f_i(w)$ at every round is often computationally costly. Hence, we take a batch of data $\mathcal{D}^i$ with respect to distribution $p_i$ to obtain an unbiased estimate $\tilde{\nabla} f_i(w, \mathcal{D}^i)$ given by

$$\tilde{\nabla} f_i(w, \mathcal{D}^i) := \frac{1}{|\mathcal{D}^i|} \sum_{(x,y) \in \mathcal{D}^i} \nabla l_i(w; x, y). \tag{6}$$

Similarly, the Hessian $\nabla^2 f_i(w)$ in (5) can be replaced by its unbiased estimate $\tilde{\nabla}^2 f_i(w, \mathcal{D}^i)$.

At round $k$ of Per-FedAvg, similar to FedAvg, first the server sends the current global model $w_k$ to a fraction of users $\mathcal{A}_k$ chosen uniformly at random with size $rn$. Each user $i \in \mathcal{A}_k$ performs $\tau$ steps of stochastic gradient descent locally and with respect to $F_i$. In particular, these local updates generate a local sequence $\{w^i_{k+1,t}\}_{t=0}^\tau$ where $w^i_{k+1,0} = w_k$ and, for $\tau \geq t \geq 1$,

$$w^i_{k+1,t} = w^i_{k+1,t-1} - \beta \tilde{\nabla} F_i(w^i_{k+1,t-1}), \tag{7}$$

where $\beta$ is the local learning rate (stepsize) and $\tilde{\nabla} F_i(w^i_{k+1,t-1})$ is an estimate of $\nabla F_i(w^i_{k+1,t-1})$ in (5). Note that the stochastic gradient $\tilde{\nabla} F_i(w^i_{k+1,t-1})$ for all local iterates is computed using independent batches $\mathcal{D}^i_t, \mathcal{D}'^i_t$, and $\mathcal{D}''^i_t$ as follows

$$\tilde{\nabla} F_i(w^i_{k+1,t-1}) := \left(I - \alpha \tilde{\nabla}^2 f_i(w^i_{k+1,t-1}, \mathcal{D}''^i_t)\right) \tilde{\nabla} f_i\left(w^i_{k+1,t-1} - \alpha \tilde{\nabla} f_i(w^i_{k+1,t-1}, \mathcal{D}^i_t), \mathcal{D}'^i_t\right). \tag{8}$$

Note that $\tilde{\nabla} F_i(w^i_{k+1,t-1})$ is a biased estimator of $\nabla F_i(w^i_{k+1,t-1})$ due to the fact that $\tilde{\nabla} f_i(w^i_{k+1,t-1} - \alpha \tilde{\nabla} f_i(w^i_{k+1,t-1}, \mathcal{D}^i_t), \mathcal{D}'^i_t)$ is a stochastic gradient that contains another stochastic gradient inside.

Once, the local updates $w^i_{k+1,\tau}$ are evaluated, users send them to the server, and the server updates its global model by averaging over the received models, i.e., $w_{k+1} = \frac{1}{rn} \sum_{i \in \mathcal{A}_k} w^i_{k+1,\tau}$.

Note that as in other MAML methods [2, 31], the update in (7) can be implemented in two stages: First, we compute $\tilde{w}^i_{k+1,t} = w^i_{k+1,t-1} - \alpha \tilde{\nabla} f_i(w^i_{k+1,t-1}, \mathcal{D}^i_t)$ and then evaluate $w^i_{k+1,t}$ by $w^i_{k+1,t} = w^{i,t-1}_{k+1} - \beta(I - \alpha \tilde{\nabla}^2 f_i(w^i_{k+1,t-1}, \mathcal{D}''^i_t)) \tilde{\nabla} f_i(\tilde{w}^i_{k+1,t}, \mathcal{D}'^i_t)$. Indeed, it can be verified the outcome of the these two steps is equivalent to the update in (7). To simplify the notation, throughout the paper, we assume that the size of $\mathcal{D}^i_t$, $\mathcal{D}'^i_t$, and $\mathcal{D}''^i_t$ is equal to $D$, $D'$, and $D''$, respectively, and for any $i$ and $t$. The steps of Per-FedAvg are depicted in Algorithm 1.

# 4 Theoretical Results

In this section, we study the convergence properties of the Personalized FedAvg (Per-FedAvg) method. We focus on nonconvex settings, and characterize the overall communication rounds between server and users to find an $\epsilon$-approximate first-order stationary point, where its formal definition follows.

**Definition 4.1.** *A random vector $w_\epsilon \in \mathbb{R}^d$ is called an $\epsilon$-approximate First-Order Stationary Point (FOSP) for problem* (3) *if it satisfies* $\mathbb{E}[\|\nabla F(w_\epsilon)\|^2] \leq \epsilon$.

**Algorithm 1:** The proposed Personalized FedAvg (Per-FedAvg) Algorithm

---

**Input:** Initial iterate $w_0$, fraction of active users $r$.
**for** $k : 0$ to $K - 1$ **do**
    Server chooses a subset of users $\mathcal{A}_k$ uniformly at random and with size $rn$;
    Server sends $w_k$ to all users in $\mathcal{A}_k$;
    **for** all $i \in \mathcal{A}_k$ **do**
        Set $w_{k+1,0}^i = w_k$;
        **for** $t : 1$ to $\tau$ **do**
            Compute the stochastic gradient $\tilde{\nabla} f_i(w_{k+1,t-1}^i, \mathcal{D}_t^i)$ using dataset $\mathcal{D}_t^i$;
            Set $\tilde{w}_{k+1,t}^i = w_{k+1,t-1}^i - \alpha \tilde{\nabla} f_i(w_{k+1,t-1}^i, \mathcal{D}_t^i)$;
            Set $w_{k+1,t}^i = w_{k+1,t-1}^i - \beta(I - \alpha \tilde{\nabla}^2 f_i(w_{k+1,t-1}^i, \mathcal{D}_t^{''i})) \tilde{\nabla} f_i(\tilde{w}_{k+1,t}^i, \mathcal{D}_t^{'i})$;
        **end for**
        Agent $i$ sends $w_{k+1,\tau}^i$ back to server;
    **end for**
    Server updates its model by averaging over received models: $w_{k+1} = \frac{1}{rn} \sum_{i \in \mathcal{A}_k} w_{k+1,\tau}^i$;
**end for**

---

Next, we formally state the assumptions required for proving our main results.

**Assumption 1.** *Functions $f_i$ are bounded below, i.e., $\min_{w \in \mathbb{R}^d} f_i(w) > -\infty$.*

**Assumption 2.** *For every $i \in \{1, \ldots, n\}$, $f_i$ is twice continuously differentiable and $L_i$-smooth, and also, its gradient is bounded by a nonnegative constant $B_i$, i.e.,*

$$\|\nabla f_i(w)\| \le B_i, \quad \|\nabla f_i(w) - \nabla f_i(u)\| \le L_i \|w - u\| \quad \forall w, u \in \mathbb{R}^d. \tag{9}$$

As we discussed in Section 3, the second-order derivative of all functions appears in the update rule of Per-FedAvg Algorithm. Hence, in the next Assumption, we impose a regularity condition on the Hessian of each $f_i$ which is also a customary assumption in the analysis of second-order methods.

**Assumption 3.** *For every $i \in \{1, \ldots, n\}$, the Hessian of function $f_i$ is $\rho_i$-Lipschitz continuous, i.e.,*

$$\|\nabla^2 f_i(w) - \nabla^2 f_i(u)\| \le \rho_i \|w - u\| \quad \forall w, u \in \mathbb{R}^d. \tag{10}$$

To simplify the analysis, in the rest of the paper, we define $B := \max_i B_i$, $L := \max_i L_i$, and $\rho := \max_i \rho_i$ which can be, respectively, considered as a bound on the norm of gradient of $f_i$, smoothness parameter of $f_i$, and Lipschitz continuity parameter of Hessian $\nabla^2 f_i$, for $i = 1, \ldots, n$.

Our next assumption provides upper bounds on the variances of gradient and Hessian estimations.

**Assumption 4.** *For any $w \in \mathbb{R}^d$, the stochastic gradient $\nabla l_i(x, y; w)$ and Hessian $\nabla^2 l_i(x, y; w)$, computed with respect to a single data point $(x, y) \in \mathcal{X}_i \times \mathcal{Y}_i$, have bounded variance, i.e.,*

$$\mathbb{E}_{(x,y) \sim p_i} \left[ \|\nabla l_i(x, y; w) - \nabla f_i(w)\|^2 \right] \le \sigma_G^2, \tag{11}$$

$$\mathbb{E}_{(x,y) \sim p_i} \left[ \|\nabla^2 l_i(x, y; w) - \nabla^2 f_i(w)\|^2 \right] \le \sigma_H^2. \tag{12}$$

Finally, we state our last assumption which characterizes the *similarity* between the tasks of users.

**Assumption 5.** *For any $w \in \mathbb{R}^d$, the gradient and Hessian of local functions $f_i(w)$ and the average function $f(w) = \sum_{i=1}^n f_i(w)$ satisfy the following conditions*

$$\frac{1}{n} \sum_{i=1}^n \|\nabla f_i(w) - \nabla f(w)\|^2 \le \gamma_G^2, \qquad \frac{1}{n} \sum_{i=1}^n \|\nabla^2 f_i(w) - \nabla^2 f(w)\|^2 \le \gamma_H^2. \tag{13}$$

Assumption 5 captures the diversity between the gradients and Hessians of users. Note that under Assumption 2, the conditions in Assumption 5 are automatically satisfied for $\gamma_G = 2B$ and $\gamma_H = 2L$. However, we state this assumption separately to highlight the role of similarity of functions corresponding to different users in convergence analysis of Per-FedAvg. In particular, in the following subsection, we highlight the connections between this assumption and the similarity of distributions $p_i$ for the case of supervised learning (2) under two different distribution distances.

## 4.1 On the Connections of Task Similarity and Distribution Distances

Recall the definition of $f_i$ in (2). Note that Assumption 5 captures the similarity of loss functions of different users. Hence, a fundamental question here is whether this has any connection with the closeness of distributions $p_i$. We study this connection by considering two different distances: Total Variation (TV) distance and 1-Wasserstein distance. Throughout this subsection, we assume all users have the same loss function $l(.;.)$ over the same set of inputs and labels, i.e., $f_i(w) := \mathbb{E}_{z \sim p_i}[l(z;w)]$ where $z := (x,y) \in \mathcal{Z} := \mathcal{X} \times \mathcal{Y}$. Also, let $p = \frac{1}{n}\sum_i p_i$ denote the average of all users' distributions.

• **Total Variation (TV) Distance:** For distributions $q_1$ and $q_2$ over countable set $\mathcal{Z}$, their TV distance is given by $\|q_1 - q_2\|_{TV} = \frac{1}{2}\sum_{z \in \mathcal{Z}}|q_1(z) - q_2(z)|$. If we assume a stronger version of Assumption 2 holds where for any $z \in \mathcal{Z}$ and $w \in \mathbb{R}^d$, we have $\|\nabla_w l(z;w)\| \leq B$ and $\|\nabla_w^2 l(z;w)\| \leq L$, then Assumption 5 holds with (check Appendix B)

$$\gamma_G^2 = 4B^2\frac{1}{n}\sum_{i=1}^{n}\|p_i - p\|_{TV}^2, \qquad \gamma_H^2 = 4L^2\frac{1}{n}\sum_{i=1}^{n}\|p_i - p\|_{TV}^2. \tag{14a}$$

This simple derivation shows that $\gamma_G$ and $\gamma_H$ exactly capture the difference between the probability distributions of the users in a heterogeneous setting.

• **1-Wasserstein Distance:** The 1-Wasserstein distance between two probability distributions $q_1$ and $q_2$ over a metric space $\mathcal{Z}$ defined as $W_1(q_1,q_2) := \inf_{q \in Q(q_1,q_2)}\int_{\mathcal{Z} \times \mathcal{Z}} d(z_1,z_2)\,\mathrm{d}q(z_1,z_2)$, where $d(.,.)$ is a distance function over metric space $\mathcal{Z}$ and $Q(q_1,q_2)$ denotes the set of all measures on $\mathcal{Z} \times \mathcal{Z}$ with marginals $q_1$ and $q_2$ on the first and second coordinate, respectively. Here, we assume all $p_i$ have bounded support (note that this assumption holds in many cases as either $\mathcal{Z}$ itself is bounded or because we normalize the data). Also, we assume that for any $w$, the gradient $\nabla_w l(z;w)$ and the Hessian $\nabla_w^2 l(z;w)$ are both Lipschitz with respect to parameter $z$ and distance $d(.,.)$, i.e,

$$\|\nabla_w l(z_1;w) - \nabla_w l(z_2;w)\| \leq L_{\mathcal{Z}}d(z_1,z_2), \quad \|\nabla_w^2 l(z_1;w) - \nabla_w^2 l(z_2;w)\| \leq \rho_{\mathcal{Z}}d(z_1,z_2). \tag{15}$$

Then, Assumption 5 holds with (check Appendix B)

$$\gamma_G^2 = L_{\mathcal{Z}}^2\frac{1}{n}\sum_{i=1}^{n}W_1(p_i,p)^2, \qquad \gamma_H^2 = \rho_{\mathcal{Z}}^2\frac{1}{n}\sum_{i=1}^{n}W_1(p_i,p)^2. \tag{16}$$

This derivation does not require Assumption 2 and holds when (15) are satisfied. Finally, consider a special case where the data distributions are homogeneous, and each $p_i$ is an empirical distribution drawn from a distribution $p_u$ with sample size $m$. In this case, we have $W_1(p_i,p_u) = \mathcal{O}(1/\sqrt{m})$ [41]. Hence, since $W_1$ is a distance, it is easy to verify that $\gamma_G, \gamma_H = \mathcal{O}(1/\sqrt{m})$[1].

## 4.2 Convergence Analysis of Per-FedAvg Algorithm

In this subsection, we derive the overall complexity of Per-FedAvg for achieving an $\epsilon$-first-order stationary point. To do so, we first prove the following intermediate result which shows that under Assumptions 2 and 3, the local meta-functions $F_i(w)$ defined in (4) and their average function $F(w) = (1/n)\sum_{i=1}^{n}F_i(w)$ are smooth.

**Lemma 4.2.** *Recall the definition of $F_i(w)$ in (4) with $\alpha \in [0,1/L]$. If Assumptions 2 and 3 hold, then $F_i$ is smooth with parameter $L_F := 4L + \alpha\rho B$. As a consequence, the average function $F(w) = (1/n)\sum_{i=1}^{n}F_i(w)$ is also smooth with parameter $L_F$.*

Assumption 4 provides upper bounds on the variances of gradient and Hessian estimation for functions $f_i$. To analyze the convergence of Per-FedAvg, however, we require upper bounds on the bias and variance of gradient estimation of $F_i$. We derive these bounds in the following lemma.

**Lemma 4.3.** *Recall the definition of the gradient estimate $\tilde{\nabla}F_i(w)$ in (8) which is computed using $\mathcal{D}$, $\mathcal{D}'$, and $D''$ that are independent batches with size $D$, $D'$, and $D''$, respectively. If Assumptions 2-4*

*hold, then for any $\alpha \in [0, 1/L]$ and $w \in \mathbb{R}^d$ we have*

$$\left\| \mathbb{E}\left[ \tilde{\nabla}F_i(w) - \nabla F_i(w)\right]\right\| \leq \frac{2\alpha L \sigma_G}{\sqrt{D}},$$

$$\mathbb{E}\left[\left\| \tilde{\nabla}F_i(w) - \nabla F_i(w)\right\|^2\right] \leq \sigma_F^2 := 12\left[B^2 + \sigma_G^2\left[\frac{1}{D'} + \frac{(\alpha L)^2}{D}\right]\right]\left[1 + \sigma_H^2 \frac{\alpha^2}{4D''}\right] - 12B^2.$$

To measure the tightness of this result, we consider two special cases. First, if the exact gradients and Hessians are available, i.e., $\sigma_G = \sigma_H = 0$, then $\sigma_F = 0$ as well which is expected as we can compute exact $\nabla F_i$. Second, for the classic federated learning problem, i.e., $\alpha = 0$ and $F_i = f_i$, we have $\sigma_F = \mathcal{O}(1)\sigma_G^2/D'$ which is tight up to constants.

Next, we use the similarity conditions for the functions $f_i$ in Assumption 5 to study the similarity between gradients of the functions $F_i$.

**Lemma 4.4.** *Recall the definition of $F_i(w)$ in (4) and assume that $\alpha \in [0, 1/L]$. Suppose that the conditions in Assumptions 2, 3, and 5 are satisfied. Then, for any $w \in \mathbb{R}^d$, we have*

$$\frac{1}{n}\sum_{i=1}^n \|\nabla F_i(w) - \nabla F(w)\|^2 \leq \gamma_F^2 := 3B^2\alpha^2\gamma_H^2 + 192\gamma_G^2.$$

To check the tightness of this result, we focus on two special cases as we did for Lemma 4.3 . First, if $\nabla f_i$ are all equal, i.e., $\gamma_G = \gamma_H = 0$, then $\gamma_F = 0$. This is indeed expected as all $\nabla F_i$ are equal to each other in this case. Second, for the classic federated learning problem, i.e., $\alpha = 0$ and $F_i = f_i$, we have $\gamma_F = \mathcal{O}(1)\gamma_G$ that is optimal up to a constant factor given the conditions in Assumption 5.

**Theorem 4.5.** *Consider the objective function $F$ defined in (3) for the case that $\alpha \in (0, 1/L]$. Suppose that the conditions in Assumptions 1-4 are satisfied, and recall the definitions of $L_F$, $\sigma_F$, and $\eta_F$ from Lemmas 4.2-4.4. Consider running Algorithm 1 for $K$ rounds with $\tau$ local updates in each round and with $\beta \leq 1/(10\tau L_F)$. Then, the following first-order stationary condition holds*

$$\frac{1}{\tau K}\sum_{k=0}^{K-1}\sum_{t=0}^{\tau-1} E\left[\|\nabla F(\bar{w}_{k+1,t})\|^2\right] \leq \frac{4(F(w_0) - F^*)}{\beta \tau K}$$

$$+ \mathcal{O}(1)\left(\beta L_F\left(1 + \beta L_F \tau(\tau-1)\right)\sigma_F^2 + \beta L_F \gamma_F^2\left(\frac{1-r}{r(n-1)} + \beta L_F \tau(\tau-1)\right) + \frac{\alpha^2 L^2 \sigma_G^2}{D}\right),$$

*where $\bar{w}_{k+1,t}$ is the average of iterates of users in $\mathcal{A}_k$ at time $t$, i.e., $\bar{w}_{k+1,t} = \frac{1}{rn}\sum_{i\in\mathcal{A}_k} w_{k+1,t}^i$, and in particular, $\bar{w}_{k+1,0} = w_k$ and $\bar{w}_{k+1,\tau} = w_{k+1}$.*

Note that $\sigma_F$ is not a constant, and as expressed in Lemma 4.3, we can make it arbitrary small by choosing batch sizes $D$, $D'$, or $D''$ large enough. To see how tight our result is, we again focus on special cases. Let $\alpha = 0$, $\tau = 1$, and $r = 1$. In this case, Per-FedAvg reduces to stochastic gradient descent, where the only source of stochasticity is the batches of gradient. In this case, the second term in the right hand side reduces to $\mathcal{O}\left(\beta L_F \sigma_F^2\right)$ where, here, $\sigma_F^2$ itself is equal to $\sigma_G^2/D$. This is the classic result for stochastic gradient descent for nonconvex functions, and we recover the lower bounds [42]. Also, it is worth noting that the term $\alpha^2 L^2 \sigma_G^2/D$ appears in the upper bound due to the fact that $\tilde{\nabla}F_i(w)$ is a *biased* estimator of $\nabla F_i(w)$. This bias term will be eliminated if we assume that we have access to the exact gradients at training time (see the discussion after Lemma 4.3), which is, for instance, the case in [35], where the authors focus on the deterministic case.

Next, we characterize the choices of $\tau$, $K$, and $\beta$ in terms of the required accuracy $\epsilon$ to obtain the best possible complexity bound for the result in Theorem 4.5.

**Corollary 4.6.** *Suppose the conditions in Theorem 4.5 are satisfied. If we set the number of local updates as $\tau = \mathcal{O}(\epsilon^{-1/2})$, number of communication rounds with the server as $K = \mathcal{O}(\epsilon^{-3/2})$, and stepsize of Per-FedAvg as $\beta = \epsilon$, then we find an $\mathcal{O}(\epsilon + \frac{\alpha^2\sigma_G^2}{D})$-first-order stationary point of $F$.*

The result in Corollary 4.6 shows that to achieve an $\mathcal{O}(\epsilon + \frac{\alpha^2\sigma_G^2}{D})$-first-order stationary point of $F$ the Per-FedAvg algorithm requires $K = \mathcal{O}(\epsilon^{-3/2})$ rounds of communication between users and the server. Indeed, by setting $D = \mathcal{O}(\epsilon^{-1})$ or setting the meta-step stepsize as $\alpha = \mathcal{O}(\epsilon^{1/2})$ Per-FedAvg can find an $\epsilon$-first-order stationary point of $F$ for any arbitrary $\epsilon > 0$.

**Remark 4.7.** *The result of Theorem 4.5 and Corollary 4.6 provide an upper bound on the average of* $E\left[\|\nabla F(\bar{w}_{k+1,t})\|^2\right]$ *for all* $k \in \{0, 1, ..., K-1\}$ *and* $t \in \{0, 1, ..., \tau-1\}$. *However, one concern here is that due to the structure of Algorithm 1, for any* $k$, *we only have access to* $\bar{w}_{k+1,t}$ *for* $t = 0$. *To address this issue, at any iteration* $k$, *the center can choose* $t_k \in \{0, 1..., \tau-1\}$ *uniformly at random, and ask all the users in* $\mathcal{A}_k$ *to send* $w^i_{k+1,t_k}$ *back to the server, in addition to* $w^i_{k+1,\tau}$. *By following this scheme we can ensure that the same upper bound also hods for the expected average models at the server, i.e.,* $\frac{1}{K} \sum_{k=0}^{K-1} E\left[\|\nabla F(\bar{w}_{k+1,t_k})\|^2\right]$.

**Remark 4.8.** *It is worth noting that it is possible to achieve the same complexity bound using a diminishing stepsize. We will further discuss this at the end of Appendix G.*

## 5 Numerical Experiments

In this section, we numerically study the role of personalization when the data distributions are heterogeneous. In particular, we consider the multi-class classification problem over MNIST [43] and CIFAR-10 [44] datasets and distribute the training data between $n$ users as follows: (i) Half of the users, *each* have $a$ images of *each* of the first five classes; (ii) The rest, *each* have $a/2$ images from only *one* of the first five classes and $2a$ images from only *one* of the other five classes (see Appendix I for an illustration). We set the parameter $a$ as $a = 196$ and $a = 68$ for MNIST and CIFAR-10 datasets, respectively. This way, we create an example where the distribution of images over all the users are different. Similarly, we divide the test data over the nodes with the same distribution as the one for the training data. Note that for this particular example in which the user's distributions are significantly different, our goal is not to achieve state-of-the-art accuracy. Rather, we aim to provide an example to compare the various approaches for obtaining personalization in the heterogenous setting. Indeed, by using more complex neural networks the results for all the considered algorithms would improve; however, their relative performance would stay the same.

We focus on three algorithms: The first method that we consider is the FedAvg method, and, to do a fair comparison, we take the output of the FedAvg method, and update it with one step of stochastic gradient descent with respect to the test data, and then evaluate its performance. The second and third algorithms that we consider are two different efficient approximations of Per-FedAvg. Similarly, we evaluate the performance of these methods for the case that one step of local stochastic gradient descent is performed during test time. To formally explain these two approximate versions of Per-FedAvg, note that the implementation of Per-FedAvg requires access to second-order information which is computationally costly. To address this issue, we consider two different approximations:

(i) First, we replace the gradient estimate with its first-order approximation which ignores the Hessian term, i.e., $\tilde{\nabla} F_i(w^i_{k+1,t-1})$ in (8) is approximated by $\tilde{\nabla} f_i(w^i_{k+1,t-1} - \alpha\tilde{\nabla} f_i(w^i_{k+1,t-1}, \mathcal{D}^i_t), \mathcal{D}'^i_t)$. This is the same idea deployed in First-Order MAML (FO-MAML) in [2], and it has been studied empirically for the federated learning setting in [32]. We refer to this algorithm as Per-FedAvg (FO).

(ii) Second, we use the idea of the HF-MAML, proposed in [31], in which the Hessian-vector product in the MAML update is replaced by difference of gradients using the following approximation: $\nabla^2\phi(w)u \approx (\nabla\phi(u + \delta v) - \nabla\phi(u - \delta v))/\delta$. We refer to this algorithm as Per-FedAvg (HF).

As shown in [31], for small stepsize at test time $\alpha$ both FO-MAML and HF-MAML perform well, but as $\alpha$ becomes large, HF-MAML outperforms FO-MAML in the centralized setting. A more detailed discussion on Per-FedAvg (FO) and Per-FedAvg (HF) is provided in Appendix H. Moreover, there we discuss how our analysis can be extended to these two methods. Note that the model obtained by any of these three methods is later updated using one step of stochastic gradient descent at the test time, and hence they have the same budget at the test time.

We use a neural network with two hidden layers with sizes 80 and 60, and we use Exponential Linear Unit (ELU) activation function. We take $n = 50$ users in the network, and run all three algorithms for $K = 1000$ rounds. At each round, we assume $rn$ agents with $r = 0.2$ are chosen to run $\tau$ local updates. The batch sizes are $D = D' = 40$ and the learning rate is $\beta = 0.001$. Part of the code is adopted from [45]. Note that the reported results for all the considered methods corresponds to the average test accuracy among all users, after running one step of local stochastic gradient descent.

The test accuracy results along with the 95% confidence intervals are reported in Table 1. For MNIST dataset, both Per-FedAvg methods achieve a marginal gain compared to FedAvg. However, the achieved gain from using Per-FedAvg (HF) compared to FedAvg is more significant for CIFAR-10

Table 1: Comparison of test accuracy of different algorithms given different parameters

| Dataset | Parameters | Algorithms | | |
|---------|-----------|-----------|-----------|-----------|
| | | FedAvg + update | Per-FedAvg (FO) | Per-FedAvg (HF) |
| MNIST | $\tau = 10, \alpha = 0.01$ | $75.96\% \pm 0.02\%$ | $78.00\% \pm 0.02\%$ | $79.85\% \pm 0.02\%$ |
| | $\tau = 4, \alpha = 0.01$ | $60.18\% \pm 0.02\%$ | $64.55\% \pm 0.02\%$ | $70.94\% \pm 0.03\%$ |
| CIFAR-10 | $\tau = 10, \alpha = 0.001$ | $40.49\% \pm 0.07\%$ | $\mathbf{46.98\% \pm 0.1\%}$ | $\mathbf{50.44\% \pm 0.15\%}$ |
| | $\tau = 4, \alpha = 0.001$ | $38.38\% \pm 0.07\%$ | $34.04\% \pm 0.08\%$ | $\mathbf{43.73\% \pm 0.11\%}$ |
| | $\tau = 4, \alpha = 0.01$ | $35.97\% \pm 0.17\%$ | $25.32\% \pm 0.18\%$ | $\mathbf{46.32\% \pm 0.12\%}$ |
| | $\tau = 4, \alpha = 0.01,$ diff. hetero. | $58.59\% \pm 0.11\%$ | $37.71\% \pm 0.23\%$ | $\mathbf{71.25\% \pm 0.05\%}$ |

dataset. In particular, we have three main observations here: (i) For $\alpha = 0.001$ and $\tau = 10$, Per-FedAvg (FO) and Per-FedAvg (HF) perform almost similarly, and better than FedAvg. In addition, decreasing $\tau$ leads to a decrease in the performance of all three algorithms, which is expected as the total number of iterations decreases. (ii) Next, we study the role of $\alpha$. By increasing $\alpha$ from 0.001 to 0.01, for $\tau = 4$, the performance of Per-FedAvg (HF) improves, which could be due to the fact that model adapts better with user data at test time. However, as discussed above, for larger $\alpha$, Per-FedAvg (FO) performance drops significantly. (iii) Third, we examine the effect of changing the level of data heterogeneity. To do so, we change the data distribution of half of the users that have $a/2$ images from one of the first five classes by removing these images from their dataset. As the last line of Table 1 shows, Per-FedAvg (HF) performs significantly better that FedAvg under these new distributions, while Per-FedAvg (FO) still suffers from the issue we discussed in (ii). In summary, the more accurate implementation of Per-FedAvg, i.e., Per-FedAvg (HF), outperforms FedAvg in all cases and leads to a more personalized solution.

## 6    Conclusion

We considered the Federated Learning (FL) problem in the heterogeneous case, and studied a *personalized* variant of the classic FL formulation in which our goal is to find a proper initialization model for the users that can be quickly adapted to the local data of each user after the training phase. We highlighted the connections of this formulation with Model-Agnostic Meta-Learning (MAML), and showed how the decentralized implementation of MAML, which we called Per-FedAvg, can be used to solve the proposed personalized FL problem. We also characterized the overall complexity of Per-FedAvg for achieving first-order optimality in nonconvex settings. Finally, we provided a set of numerical experiments to illustrate the performance of two different first-order approximations of Per-FedAvg and their comparison with the FedAvg method, and showed that the solution obtained by Per-FedAvg leads to a more personalized solution compared to the solution of FedAvg.

## Broader Impact

Federated Learning (FL) provides a framework for training machine learning models efficiently and in a distributed manner. Due to these favorable properties, it has gained significant attention and has been deployed in a broad range of applications with critical societal benefits. These applications go from healthcare systems, where machine learning models can be trained while preserving patients' privacy, to image classification and NLP models, where tech companies can improve their neural networks without requiring users to share their data with a server or other users. In our work, we study one of the challenges in FL, which is the personalization aspect. The main question that we try to answer from a theoretical point of view is whether we can have a user-oriented variant of classic FL algorithms that can adapt to each user data while enjoying the distributed architecture of FL. We show the answer is positive, and provide rigorous theoretical guarantees for algorithms that can be used in all applications mentioned above to achieve more personalized models in FL framework. Indeed, this result could have a broad impact on improving the quality of users' models in several applications that deploy federated learning such as healthcare systems.

## Acknowledgments and Disclosure of Funding

Research was sponsored by the United States Air Force Research Laboratory and was accomplished under Cooperative Agreement Number FA8750-19-2-1000. The views and conclusions contained in this document are those of the authors and should not be interpreted as representing the official policies, either expressed or implied, of the United States Air Force or the U.S. Government. The U.S. Government is authorized to reproduce and distribute reprints for Government purposes notwithstanding any copyright notation herein. Alireza Fallah acknowledges support from MathWorks Engineering Fellowship. The research of Aryan Mokhtari is supported by NSF Award CCF-2007668.

## Footnotes

[1]While our focus here is to elaborate on the dependence of Wasserstein distance on the number of samples, it is worth noting that one drawback of this bound is that the convergence speed of Wasserstein distance in dimension is exponentially slow.

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
