[Supplementary Material]

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

# Supplementary Material

## A  Intermediate Notes

Note that the gradient Lipschitz assumption, i.e., the second inequality in (9), also implies that $f_i$ satisfies the following conditions for all $w, u \in \mathbb{R}^d$:

$$- L_i I_d \preceq \nabla^2 f_i(w) \preceq L_i I_d, \tag{17a}$$

$$\mid f_i(w) - f_i(u) - \nabla f_i(u)^\top (w - u)\mid \leq \frac{L_i}{2}\|w - u\|^2. \tag{17b}$$

## B  Proofs of results in Subsection 4.1

### B.1  TV Distance

Note that

$$
\begin{aligned}
\|\nabla f_i(w) - \nabla f(w)\| &= \left\| \sum_{z \in \mathcal{Z}} \nabla_w l(z; w)\, (p_i(z) - p(z)) \right\| \\
&\leq \sum_{z \in \mathcal{Z}} \|\nabla_w l(z; w)\|\, |p_i(z) - p(z)| \\
&\leq B \sum_{z \in \mathcal{Z}} |p_i(z) - p(z)| = 2B\|p_i - p\|_{TV}
\end{aligned}
\tag{18}
$$

where the second inequality holds due to the assumption that $\|\nabla_w l(z; w)\| \leq B$ for any $w$ and $z$. Plugging (18) in $\frac{1}{n}\sum_{i=1}^n \|\nabla f_i(w) - \nabla f(w)\|^2$, gives us the desired result. The other result on Hessians can be proved similarly.

### B.2  1-Wasserstein Distance

We claim that for any $i$ and $w \in \mathbb{R}^d$, we have

$$\|\nabla f_i(w) - \nabla f(w)\| \leq L_{\mathcal{Z}} W_1(p_i, p), \tag{19}$$

which will immediately give us one of the two results. To show this, first, note that

$$
\begin{aligned}
\|\nabla f_i(w) - \nabla f(w)\| &= \sup_{v \in \mathbb{R}^d : \|v\| \leq 1} v^\top (\nabla f_i(w) - \nabla f(w)) \\
&= \sup_{v \in \mathbb{R}^d : \|v\| \leq 1} \left( \mathbb{E}_{z \sim p_i}\left[ v^\top \nabla l(z; w) \right] - \mathbb{E}_{z \sim p}\left[ v^\top \nabla l(z; w) \right] \right)
\end{aligned}
$$

Thus, we need to show for any $v \in \mathbb{R}^d$ with $\|v\| \leq 1$, we have

$$\mathbb{E}_{z \sim p_i}\left[ v^\top \nabla l(z; w) \right] - \mathbb{E}_{z \sim p}\left[ v^\top \nabla l(z; w) \right] \leq L_{\mathcal{Z}} W_1(p_i, p). \tag{20}$$

Next, note that since $p_i$ and $p$ both have bounded support, by Kantorovich-Rubinstein Duality [46], we have

$$W_1(p_i, p) = \sup\left\{ \mathbb{E}_{z \sim p_i}\left[ g(z) \right] - \mathbb{E}_{z \sim p}\left[ g(z) \right] \mid \text{ continuous } g : \mathcal{Z} \to \mathbb{R}, \mathrm{Lip}(g) \leq 1 \right\}. \tag{21}$$

Using this result, to show (20), it suffices to show $g(z) = v^\top \nabla l(z; w)$ is $L_{\mathcal{Z}}$-Lipschitz. Note that Cauchy-Schwarz inequality implies

$$\|v^\top \nabla l(z_1; w) - v^\top \nabla l(z_2; w)\| \leq \|v\| \|\nabla l(z_1; w) - \nabla l(z_2; w)\| \leq L_{\mathcal{Z}} d(z_1, z_2) \tag{22}$$

where the last inequality is obtained using $\|v\| \leq 1$ along with (15).

Finally, note that we can similarly show the result for $\gamma_H$ by considering the fact that

$$
\begin{aligned}
\|\nabla^2 f_i(w) - \nabla^2 f(w)\| &= \max_{\xi \in \{1, -1\}} \sup_{v \in \mathbb{R}^d : \|v\| \leq 1} \xi v^\top \left( \nabla^2 f_i(w) - \nabla^2 f(w) \right) v \\
&= \max_{\xi \in \{1, -1\}} \sup_{v \in \mathbb{R}^d : \|v\| \leq 1} \xi \left( \mathbb{E}_{z \sim p_i}\left[ v^\top \nabla^2 l(z; w) v \right] - \mathbb{E}_{z \sim p}\left[ v^\top \nabla^2 l(z; w) v \right] \right)
\end{aligned}
$$

and taking the functions $g(z) = v^\top \nabla^2 l(z; w) v$ and $g(z) = -v^\top \nabla^2 l(z; w) v$ along with using Kantorovich-Rubinstein Duality Theorem again.

## C  Proof of Lemma 4.2

Recall that
$$\nabla F_i(w) = \left(I - \alpha \nabla^2 f_i(w)\right) \nabla f_i(w - \alpha \nabla f_i(w)). \tag{23}$$

Given this, note that

$$
\begin{aligned}
&\|\nabla F_i(w_1) - \nabla F_i(w_2)\| \\
&= \left\| \left(I - \alpha \nabla^2 f_i(w_1)\right) \nabla f_i(w_1 - \alpha \nabla f_i(w_1)) - \left(I - \alpha \nabla^2 f_i(w_2)\right) \nabla f_i(w_2 - \alpha \nabla f_i(w_2)) \right\| \\
&= \left\| \left(I - \alpha \nabla^2 f_i(w_1)\right) \left(\nabla f_i(w_1 - \alpha \nabla f_i(w_1)) - \nabla f_i(w_2 - \alpha \nabla f_i(w_2))\right) \right. \\
&\quad \left. + \left(\left(I - \alpha \nabla^2 f_i(w_1)\right) - \left(I - \alpha \nabla^2 f_i(w_2)\right)\right) \nabla f_i(w_2 - \alpha \nabla f_i(w_2)) \right\| \\
&\leq \left\| I - \alpha \nabla^2 f_i(w_1) \right\| \left\| \nabla f_i(w_1 - \alpha \nabla f_i(w_1)) - \nabla f_i(w_2 - \alpha \nabla f_i(w_2)) \right\| \\
&\quad + \alpha \left\| \nabla^2 f_i(w_1) - \nabla^2 f_i(w_2) \right\| \left\| \nabla f_i(w_2 - \alpha \nabla f_i(w_2)) \right\|
\end{aligned}
\tag{24}
$$
$$\tag{25}$$

where (24) is obtained by adding and subtracting $\left(I - \alpha \nabla^2 f_i(w_1)\right) \nabla f_i(w_2 - \alpha \nabla f_i(w_2))$ and the last inequality follows from the triangle inequality and the definition of matrix norm. Now, we bound two terms of (25) separately.

First, note that by (17a), $\left\| I - \alpha \nabla^2 f_i(w_1) \right\| \leq 1 + \alpha L$. Using this along with smoothness of $f_i$, we have

$$
\begin{aligned}
&\left\| I - \alpha \nabla^2 f_i(w_1) \right\| \left\| \nabla f_i(w_1 - \alpha \nabla f_i(w_1)) - \nabla f_i(w_2 - \alpha \nabla f_i(w_2)) \right\| \\
&\leq (1 + \alpha L) L \left\| w_1 - \alpha \nabla f_i(w_1)) - w_2 + \alpha \nabla f_i(w_2) \right\| \\
&\leq (1 + \alpha L) L \left( \| w_1 - w_2 \| + \alpha \| \nabla f_i(w_1) - \nabla f_i(w_2) \| \right) \\
&\leq (1 + \alpha L) L (1 + \alpha L) \| w_1 - w_2 \| \\
&\leq 4L \| w_1 - w_2 \|,
\end{aligned}
\tag{26}
$$

where we used smoothness of $f_i$ along with $\alpha \leq 1/L$.

For the second term, Using (9) in Assumption 2 along with Assumption 3 implies
$$\alpha \left\| \nabla^2 f_i(w_1) - \nabla^2 f_i(w_2) \right\| \left\| \nabla f_i(w_2 - \alpha \nabla f_i(w_2)) \right\| \leq \alpha \rho B \| w_1 - w_2 \|. \tag{27}$$
Putting (26) and (27) together, we obtain the desired result.

## D  Proof of Lemma 4.3

Recall that the expression for the stochastic gradient $\tilde{\nabla} F_i(w)$ is given by
$$\tilde{\nabla} F_i(w) = \left(I - \alpha \tilde{\nabla}^2 f_i(w, \mathcal{D}'')\right) \tilde{\nabla} f_i\left(w - \alpha \tilde{\nabla} f_i(w, \mathcal{D}), \mathcal{D}'\right) \tag{28}$$
which can be written as
$$\tilde{\nabla} F_i(w) = \left(I - \alpha \nabla^2 f_i(w) + e_1\right) \left(\nabla f_i\left(w - \alpha \nabla f_i(w)\right) + e_2\right). \tag{29}$$
Note that in the above expression $e_1$ and $e_2$ are given by
$$e_1 = \alpha \left(\nabla^2 f_i(w) - \tilde{\nabla}^2 f_i(w, \mathcal{D}'')\right),$$
and
$$e_2 = \tilde{\nabla} f_i(w - \alpha \tilde{\nabla} f_i(w, \mathcal{D}), \mathcal{D}') - \nabla f_i\left(w - \alpha \nabla f_i(w)\right).$$

Based on Assumption 4, it can be easily shown that
$$\mathbb{E}\left[e_1\right] = 0, \tag{30a}$$
$$\mathbb{E}\left[\|e_1\|^2\right] \leq \alpha^2 \frac{\sigma_H^2}{D''}. \tag{30b}$$

Next, we proceed to bound the first and second moments of $e_2$. To do so, first note that $e_2$ can also be written as
$$
\begin{aligned}
e_2 &= \left(\tilde{\nabla} f_i\left(w - \alpha \tilde{\nabla} f_i(w, \mathcal{D}), \mathcal{D}'\right) - \nabla f_i\left(w - \alpha \tilde{\nabla} f_i(w, \mathcal{D})\right)\right) \\
&\quad + \left(\nabla f_i\left(w - \alpha \tilde{\nabla} f_i(w, \mathcal{D})\right) - \nabla f_i\left(w - \alpha \nabla f_i(w)\right)\right).
\end{aligned}
\tag{31}
$$

Note that, conditioning on $\mathcal{D}$, the first term is zero mean and the second term is deterministic. Therefore,

$$\|\mathbb{E}[e_2]\| = \left\|\mathbb{E}\left[\nabla f_i\left(w - \alpha\tilde{\nabla} f_i(w,\mathcal{D})\right) - \nabla f_i\left(w - \alpha\nabla f_i(w)\right)\right]\right\|$$

$$\leq \mathbb{E}\left[\left\|\nabla f_i\left(w - \alpha\tilde{\nabla} f_i(w,\mathcal{D})\right) - \nabla f_i\left(w - \alpha\nabla f_i(w)\right)\right\|\right]$$

$$\leq \alpha L \mathbb{E}\left[\left\|\tilde{\nabla} f_i(w,\mathcal{D}) - \nabla f_i(w)\right\|\right] \tag{32}$$

$$\leq \frac{\alpha L \sigma_G}{\sqrt{D}}, \tag{33}$$

where (32) is obtained using smoothness of $f_i$. The last inequality is also obtained using

$$\mathbb{E}\left[\left\|\tilde{\nabla} f_i(w,\mathcal{D}) - \nabla f_i(w)\right\|^2\right] \leq \frac{\sigma_G^2}{D}. \tag{34}$$

In addition, we have

$$\mathbb{E}\left[\|e_2\|^2\right] = \mathbb{E}\left[\mathbb{E}\left[\|e_2\|^2|\mathcal{D}\right]\right]$$

$$= \mathbb{E}\left[\left\|\tilde{\nabla} f_i\left(w - \alpha\tilde{\nabla} f_i(w,\mathcal{D}),\mathcal{D}'\right) - \nabla f_i\left(w - \alpha\tilde{\nabla} f_i(w,\mathcal{D})\right)\right\|^2\right]$$

$$+ \mathbb{E}\left[\left\|\nabla f_i\left(w - \alpha\tilde{\nabla} f_i(w,\mathcal{D})\right) - \nabla f_i\left(w - \alpha\nabla f_i(w)\right)\right\|^2\right]$$

$$\leq \frac{\sigma_G^2}{D'} + L^2\alpha^2\mathbb{E}\left[\left\|\tilde{\nabla} f_i(w,\mathcal{D}) - \nabla f_i(w)\right\|^2\right] \tag{35}$$

$$\leq \sigma_G^2\left(\frac{1}{D'} + \frac{(\alpha L)^2}{D}\right) \tag{36}$$

where (36) follows from (34), and (35) is obtained using smoothness of $f_i$ along with the fact that

$$\mathbb{E}\left[\left\|\tilde{\nabla} f_i\left(w - \alpha\tilde{\nabla} f_i(w,\mathcal{D}),\mathcal{D}'\right) - \nabla f_i\left(w - \alpha\tilde{\nabla} f_i(w,\mathcal{D})\right)\right\|^2\right] \leq \frac{\sigma_G^2}{D'}.$$

Next, note that, by comparing (29) and (5), along with the fact that $e_1$ and $e_2$ are independent, and $e_1$ is zero-mean (30a), we have

$$\mathbb{E}\left[\tilde{\nabla} F_i(w) - \nabla F_i(w)\right] = (I - \alpha\nabla^2 f_i(w))\mathbb{E}[e_2]. \tag{37}$$

Hence, by taking the norm of both sides, we obtain

$$\left\|\mathbb{E}\left[\tilde{\nabla} F_i(w) - \nabla F_i(w)\right]\right\| = \left\|(I - \alpha\nabla^2 f_i(w))\mathbb{E}[e_2]\right\|$$

$$\leq \left\|(I - \alpha\nabla^2 f_i(w))\right\|\|\mathbb{E}[e_2]\| \tag{38}$$

where the last inequality follows from the definition of matrix norm. Now, using (33) along with the fact that $\|I - \alpha\nabla^2 f_i(w)\| \leq 1 + \alpha L \leq 2$ gives us the first result in Lemma 4.3.

To show the other result, note that, by comparing (29) and (5), along with the matrix norm definition, we have

$$\left\|\tilde{\nabla} F_i(w) - \nabla F_i(w)\right\| \leq \|I - \alpha\nabla^2 f_i(w)\|\|e_2\| + \|e_1\|\|\nabla f_i\left(w - \alpha\nabla f_i(w)\right)\| + \|e_1\|\|e_2\|. \tag{39}$$

As a result, by the Cauchy-Schwarz inequality $(a + b + c)^2 \leq 3(a^2 + b^2 + c^2)$ for $a, b, c \geq 0$, we have

$$\left\|\tilde{\nabla} F_i(w) - \nabla F_i(w)\right\|^2$$

$$\leq 3\|I - \alpha\nabla^2 f_i(w)\|^2\|e_2\|^2 + 3\|e_1\|^2\|\nabla f_i\left(w - \alpha\nabla f_i(w)\right)\|^2 + 3\|e_1\|^2\|e_2\|^2. \tag{40}$$

By taking expectation, and using the fact that $\|I - \alpha\nabla^2 f_i(w)\| \leq 1 + \alpha L \leq 2$ and

$$\|\nabla f_i\left(w - \alpha\nabla f_i(w)\right)\| \leq B,$$

we have

$$\mathbb{E}\left[\left\|\tilde{\nabla} F_i(w) - \nabla F_i(w)\right\|^2\right] \leq 3B^2\mathbb{E}\left[\|e_1\|^2\right] + 12\mathbb{E}\left[\|e_2\|^2\right] + 3\mathbb{E}\left[\|e_1\|^2\right]\mathbb{E}\left[\|e_2\|^2\right] \tag{41}$$

where we also used the fact that $e_1$ and $e_2$ are independent as $\mathcal{D}''$ is independent from $\mathcal{D}$ and $\mathcal{D}'$. Plugging (30b) and (36) in (41), we obtain

$$\mathbb{E}\left[\left\|\tilde{\nabla}F_i(w) - \nabla F_i(w)\right\|^2\right]$$

$$\leq 3B^2\alpha^2\frac{\sigma_H^2}{D''} + 12\sigma_G^2\left(\frac{1}{D'} + \frac{(\alpha L)^2}{D}\right) + 3\alpha^2\sigma_G^2\sigma_H^2\left(\frac{1}{D'D''} + \frac{(\alpha L)^2}{DD''}\right)$$

which gives us the desired result.

# E  Proof of Lemma 4.4

Recall that

$$\nabla F_i(w) = \left(I - \alpha\nabla^2 f_i(w)\right)\nabla f_i(w - \alpha\nabla f_i(w)). \tag{42}$$

which can be expressed as

$$\nabla F_i(w) = \left(I - \alpha\nabla^2 f(w) + E_i\right)\left(\nabla f(w - \alpha\nabla f(w)) + r_i\right) \tag{43}$$

where

$$E_i = \alpha\left(\nabla^2 f(w) - \nabla^2 f_i(w)\right), \tag{44}$$

$$r_i = \nabla f_i(w - \alpha\nabla f_i(w)) - \nabla f(w - \alpha\nabla f(w)). \tag{45}$$

First, note that, by Assumption 5, we have

$$\frac{1}{n}\sum_{i=1}^{n}\|E_i\|^2 = \alpha^2\gamma_H^2. \tag{46}$$

Second, note that

$$\|r_i\| \leq \|\nabla f_i(w - \alpha\nabla f_i(w)) - \nabla f_i(w - \alpha\nabla f(w))\|$$
$$+ \|\nabla f_i(w - \alpha\nabla f(w)) - \nabla f(w - \alpha\nabla f(w))\|$$
$$\leq \alpha L\|\nabla f_i(w) - \nabla f(w)\| + \|\nabla f_i(w - \alpha\nabla f(w)) - \nabla f(w - \alpha\nabla f(w))\| \tag{47}$$

where the last inequality is obtained using (9) in Assumption 2. Now, by using $(a+b)^2 \leq 2(a^2+b^2)$, we have

$$\frac{1}{n}\sum_{i=1}^{n}\|r_i\|^2$$

$$\leq \frac{2}{n}\sum_{i=1}^{n}\left((\alpha L)^2\|\nabla f_i(w) - \nabla f(w)\|^2 + \|\nabla f_i(w - \alpha\nabla f(w)) - \nabla f(w - \alpha\nabla f(w))\|^2\right)$$

$$\leq 2\left(1 + (\alpha L)^2\right)\left(\gamma_G^2 + \gamma_G^2\right) \tag{48}$$

$$\leq 8\gamma_G^2. \tag{49}$$

where the second inequality follows from Assumption 5 and the last inequality is obtained using $\alpha L \leq 1$. Next, recall that the goal is to bound the variance of $\nabla F_i(w)$ when $i$ is drawn from a uniform distribution. We know that by subtracting a constant from a random variable, its variance does not change. Thus, variance of $\nabla F_i(w)$ is equal to variance of $\nabla F_i(w) - \left(I - \alpha\nabla^2 f(w)\right)\nabla f(w - \alpha\nabla f(w))$. Also, the variance of the latter is bounded by its second moment, and hence,

$$\frac{1}{n}\sum_{i=1}^{n}\|\nabla F_i(w) - \nabla F(w)\|^2 \leq \frac{1}{n}\sum_{i=1}^{n}\left\|E_i\nabla f(w - \alpha\nabla f(w)) + \left(I - \alpha\nabla^2 f(w)\right)r_i + E_i r_i\right\|^2$$

$$\leq \frac{1}{n}\sum_{i=1}^{n}\left(\|E_i\nabla f(w - \alpha\nabla f(w))\| + \left\|\left(I - \alpha\nabla^2 f(w)\right)r_i\right\| + \|E_i r_i\|\right)^2 \tag{50}$$

Therefore, using $\|\nabla f(w - \alpha\nabla f(w))\| \leq B$ along with $\|I - \alpha\nabla^2 f(w)\| \leq 1 + \alpha L$ and Cauchy-Schwarz inequality $(a + b + c)^2 \leq 3(a^2 + b^2 + c^2)$ for $a, b, c \geq 0$, we obtain

$$\frac{1}{n}\sum_{i=1}^{n}\|\nabla F_i(w) - \nabla F(w)\|^2 \leq 3\left(B^2\frac{1}{n}\sum_{i=1}^{n}\|E_i\|^2 + (1 + \alpha L)^2\frac{1}{n}\sum_{i=1}^{n}\|r_i\|^2 + \frac{1}{n}\sum_{i=1}^{n}\|E_ir_i\|^2\right)$$

$$\leq 3\left(B^2\frac{1}{n}\sum_{i=1}^{n}\|E_i\|^2 + 4\frac{1}{n}\sum_{i=1}^{n}\|r_i\|^2 + \frac{1}{n}\sum_{i=1}^{n}\|E_i\|^2\|r_i\|^2\right) \quad (51)$$

where the last inequality is obtained using $\alpha L \leq 1$ along with $\|E_ir_i\| \leq \|E_i\|\|r_i\|$ which comes from the definition of matrix norm. Finally, to complete the proof, notice that we have

$$\frac{1}{n}\sum_{i=1}^{n}\|E_i\|^2\|r_i\|^2 \leq \max_i\|E_i\|^2\left(\frac{1}{n}\sum_{i=1}^{n}\|r_i\|^2\right) \quad (52)$$

$$\leq \max_i\|E_i\|^2(8\gamma_G^2) \quad (53)$$

$$\leq 32(\alpha L)^2\gamma_G^2 \leq 32\gamma_G^2 \quad (54)$$

where (53) follows from (49) and the last line is obtained using $\alpha L \leq 1$ along with the fact that $\|\nabla^2 f_i(w)\| \leq L$, and thus,

$$\frac{\|E_i\|}{\alpha} = \|\nabla^2 f(w) - \nabla^2 f_i(w)\| \leq 2L. \quad (55)$$

Plugging (53) in (51) along with (46) and (49), we obtain the desired result.

## F   An Intermediate Result

**Proposition F.1.** *Recall from Section 3 that at any round $k \geq 1$, and for any agent $i \in \{1, .., n\}$, we can define a sequence of local updates $\{w_{k,t}^i\}_{t=0}^{\tau}$ where $w_{k,0}^i = w_{k-1}$ and, for $\tau \geq t \geq 1$,*

$$w_{k,t}^i = w_{k,t-1}^i - \beta\tilde{\nabla}F_i(w_{k,t-1}^i). \quad (56)$$

*We further define the average of these local updates at round $k$ and time $t$ as $w_{k,t} = 1/n\sum_{i=1}^{n}w_{k,t}^i$. Suppose that the conditions in Assumptions 2-4 are satisfied. Then, for any $\alpha \in [0, 1/L]$ and any $t \geq 0$, we have*

$$\mathbb{E}\left[\frac{1}{n}\sum_{i=1}^{n}\|w_{k,t}^i - w_{k,t}\|\right] \leq 2\beta t(1 + 2\beta L_F)^{t-1}(\sigma_F + \gamma_F), \quad (57a)$$

$$\mathbb{E}\left[\frac{1}{n}\sum_{i=1}^{n}\|w_{k,t}^i - w_{k,t}\|^2\right] \leq 4\beta^2(1 + \frac{1}{\phi})t\left(1 + \phi + 16(1 + \frac{1}{\phi})\beta^2 L_F^2\right)^{t-1}(2\sigma_F^2 + \gamma_F^2) \quad (57b)$$

*where $\phi > 0$ is an arbitrary positive constant and $L_F$, $\sigma_F$, and $\gamma_F$ are given in Lemmas 4.2, 4.3, and 4.4, respectively.*

Before stating the proof, note that an immediate consequence of this result is the following corollary:

**Corollary F.2.** *Under the same assumptions as Proposition F.1, and for any $\beta \leq 1/(10\tau L_F)$, we have*

$$\mathbb{E}\left[\frac{1}{n}\sum_{i=1}^{n}\|w_{k,t}^i - w_{k,t}\|\right] \leq 4\beta t(\sigma_F + \gamma_F), \quad (58a)$$

$$\mathbb{E}\left[\frac{1}{n}\sum_{i=1}^{n}\|w_{k,t}^i - w_{k,t}\|^2\right] \leq 35\beta^2 t\tau(2\sigma_F^2 + \gamma_F^2) \quad (58b)$$

*for any $0 \leq t \leq \tau$.*

*Proof.* Let

$$S_t := \frac{1}{n}\sum_{i=1}^{n}\mathbb{E}\left[\|w_{k,t}^i - w_{k,t}\|\right] \quad (59)$$

where $S_0 = 0$ since $w_{k,0}^i = w_{k-1}$ for any $i$. Note that

$$
\begin{aligned}
S_{t+1} &= \frac{1}{n}\sum_{i=1}^{n}\mathbb{E}\left[\|w_{k,t+1}^i - w_{k,t+1}\|\right] \\
&= \frac{1}{n}\sum_{i=1}^{n}\mathbb{E}\left[\left\|w_{k,t}^i - \beta\tilde{\nabla}F_i(w_{k,t}^i) - \frac{1}{n}\sum_{j=1}^{n}\left(w_{k,t}^j - \beta\tilde{\nabla}F_j(w_{k,t}^j)\right)\right\|\right] \\
&\leq \frac{1}{n}\sum_{i=1}^{n}\mathbb{E}\left[\|w_{k,t}^i - \frac{1}{n}\sum_{j=1}^{n}w_{k,t}^j\|\right] + \beta\frac{1}{n}\sum_{i=1}^{n}\mathbb{E}\left[\|\tilde{\nabla}F_i(w_{k,t}^i) - \frac{1}{n}\sum_{j=1}^{n}\tilde{\nabla}F_j(w_{k,t}^j)\|\right].
\end{aligned}
$$
(60)

Note that the first term in (60) is in fact $S_t$ and the second one can be upper bounded as follows

$$
\begin{aligned}
\frac{1}{n}\sum_{i=1}^{n}&\mathbb{E}\left[\|\tilde{\nabla}F_i(w_{k,t}^i) - \frac{1}{n}\sum_{j=1}^{n}\tilde{\nabla}F_j(w_{k,t}^j)\|\right] \\
&\leq \frac{1}{n}\sum_{i=1}^{n}\mathbb{E}\left[\|\nabla F_i(w_{k,t}^i) - \frac{1}{n}\sum_{j=1}^{n}\nabla F_j(w_{k,t}^j)\|\right] + \frac{1}{n}\sum_{i=1}^{n}\mathbb{E}\left[\|\nabla F_i(w_{k,t}^i) - \tilde{\nabla}F_i(w_{k,t}^i)\|\right] \\
&\quad + \frac{1}{n}\sum_{i=1}^{n}\mathbb{E}\left[\frac{1}{n}\sum_{j=1}^{n}\|\nabla F_j(w_{k,t}^j) - \tilde{\nabla}F_j(w_{k,t}^j)\|\right] \\
&\leq \frac{1}{n}\sum_{i=1}^{n}\mathbb{E}\left[\|\nabla F_i(w_{k,t}^i) - \frac{1}{n}\sum_{j=1}^{n}\nabla F_j(w_{k,t}^j)\|\right] + 2\beta\sigma_F
\end{aligned}
$$

where the last inequality is obtained using Lemma 4.3. By substituting this in (60), we obtain

$$
S_{t+1} \leq S_t + 2\beta\sigma_F + \beta\frac{1}{n}\sum_{i=1}^{n}\mathbb{E}\left[\|\nabla F_i(w_{k,t}^i) - \frac{1}{n}\sum_{j=1}^{n}\nabla F_j(w_{k,t}^j)\|\right].
$$
(61)

If we define $\eta_i := \nabla F_i(w_{k,t}^i) - \nabla F_i(w_{k,t})$, using (61), we obtain

$$
\begin{aligned}
S_{t+1} &\leq S_t + 2\beta\sigma_F + \beta\frac{1}{n}\sum_{i=1}^{n}\mathbb{E}\left[\|\nabla F_i(w_{k,t}) - \frac{1}{n}\sum_{j=1}^{n}\nabla F_j(w_{k,t})\|\right] \\
&\quad + \beta\frac{1}{n}\sum_{i=1}^{n}\mathbb{E}\left[\|\eta_i - \frac{1}{n}\sum_{j=1}^{n}\eta_j\|\right].
\end{aligned}
$$
(62)

Note that, by Lemma 4.2,

$$
\|\eta_i\| \leq L_F\|w_{k,t}^i - w_{k,t}\|,
$$
(63)

and thus,

$$
\frac{1}{n}\sum_{i=1}^{n}\|\eta_i\| \leq L_F S_t.
$$
(64)

As a result, and by using (62), we have

$$
S_{t+1} \leq (1 + 2\beta L_F)S_t + 2\beta\sigma_F + \beta\frac{1}{n}\sum_{i=1}^{n}\mathbb{E}\left[\|\nabla F_i(w_{k,t}) - \frac{1}{n}\sum_{j=1}^{n}\nabla F_j(w_{k,t})\|\right].
$$
$$
\leq (1 + 2\beta L_F)S_t + 2\beta(\sigma_F + \gamma_F)
$$
(65)

where the last inequality is obtained using Lemma 4.4. Using (65) recursively, we obtain

$$
S_{t+1} \leq \left(\sum_{j=0}^{t}(1 + 2\beta L_F)^j\right)2\beta(\sigma_F + \gamma_F) \leq 2\beta(t+1)(1 + 2\beta L_F)^t(\sigma_F + \gamma_F)
$$
(66)

which completes the proof of (57a). To prove (57b), let

$$\Sigma_t := \frac{1}{n} \sum_{i=1}^{n} \mathbb{E}\left[\|w_{k,t}^i - w_{k,t}\|^2\right]. \tag{67}$$

Similarly $\Sigma_0 = 0$. Note that

$$\Sigma_{t+1} = \frac{1}{n} \sum_{i=1}^{n} \mathbb{E}\left[\|w_{k,t+1}^i - w_{k,t+1}\|^2\right]$$

$$= \frac{1}{n} \sum_{i=1}^{n} \mathbb{E}\left[\left\|w_{k,t}^i - \beta\tilde{\nabla}F_i(w_{k,t}^i) - \frac{1}{n}\sum_{j=1}^{n}\left(w_{k,t}^j - \beta\tilde{\nabla}F_j(w_{k,t}^j)\right)\right\|^2\right]$$

$$\leq \frac{1+\phi}{n} \sum_{i=1}^{n} \mathbb{E}\left[\|w_{k,t}^i - \frac{1}{n}\sum_{j=1}^{n}w_{k,t}^j\|^2\right]$$

$$+ \beta^2 \frac{1+1/\phi}{n} \sum_{i=1}^{n} \mathbb{E}\left[\|\tilde{\nabla}F_i(w_{k,t}^i) - \frac{1}{n}\sum_{j=1}^{n}\tilde{\nabla}F_j(w_{k,t}^j)\|^2\right] \tag{68}$$

$$\leq (1+\phi)\Sigma_t + \beta^2 \frac{1+1/\phi}{n} \sum_{i=1}^{n} \mathbb{E}\left[\|\tilde{\nabla}F_i(w_{k,t}^i) - \frac{1}{n}\sum_{j=1}^{n}\tilde{\nabla}F_j(w_{k,t}^j)\|^2\right] \tag{69}$$

where (68) is obtained using $\|a+b\|^2 \leq (1+\phi)\|a\|^2 + (1+1/\phi)\|b\|^2$ for any arbitrary positive real number $\phi$. To bound the second term in (69), note that

$$\mathbb{E}\left[\|\tilde{\nabla}F_i(w_{k,t}^i) - \frac{1}{n}\sum_{j=1}^{n}\tilde{\nabla}F_j(w_{k,t}^j)\|^2\right] \leq 2\mathbb{E}\left[\|\nabla F_i(w_{k,t}^i) - \frac{1}{n}\sum_{j=1}^{n}\nabla F_j(w_{k,t}^j)\|^2\right]$$

$$+ 2\mathbb{E}\left[\left\|\left(\tilde{\nabla}F_i(w_{k,t}^i) - \nabla F_i(w_{k,t}^i)\right) + \frac{1}{n}\sum_{j=1}^{n}\left(\nabla F_j(w_{k,t}^j) - \tilde{\nabla}F_j(w_{k,t}^j)\right)\right\|^2\right]. \tag{70}$$

Now, we bound the second term in (70). Using Cauchy-Schwarz inequality

$$\left\|\sum_{l=1}^{n+1} a_l b_l\right\|^2 \leq \left(\sum_{l=1}^{n+1}\|a_l\|^2\right)\left(\sum_{l=1}^{n+1}\|b_l\|^2\right) \tag{71}$$

with $a_1 = \tilde{\nabla}F_i(w_{k,t}^i) - \nabla F_i(w_{k,t}^i), b_1 = 1$ and $a_l = 1/\sqrt{n}\,(\tilde{\nabla}F_{l-1}(w_{k,t}^{l-1}) - \nabla F_{l-1}(w_{k,t}^{l-1})), b_l = 1/\sqrt{n}$, for $l = 2, ..., n+1$, implies

$$\mathbb{E}\left[\left\|\left(\tilde{\nabla}F_i(w_{k,t}^i) - \nabla F_i(w_{k,t}^i)\right) + \frac{1}{n}\sum_{j=1}^{n}\left(\nabla F_j(w_{k,t}^j) - \tilde{\nabla}F_j(w_{k,t}^j)\right)\right\|^2\right]$$

$$\leq 2\mathbb{E}\left[\left\|\tilde{\nabla}F_i(w_{k,t}^i) - \nabla F_i(w_{k,t}^i)\right\|^2 + \frac{1}{n}\sum_{j=1}^{n}\left\|\nabla F_j(w_{k,t}^j) - \tilde{\nabla}F_j(w_{k,t}^j)\right\|^2\right]$$

$$\leq 4\sigma_F^2 \tag{72}$$

where the last inequality is obtained using Lemma 4.3. Plugging (72) in (70) and using (69), we obtain

$$\Sigma_{t+1} \leq (1+\phi)\Sigma_t + 8(1+\frac{1}{\phi})\beta^2\sigma_F^2$$

$$+ 2(1+\frac{1}{\phi})\beta^2\frac{1}{n}\sum_{i=1}^{n}\mathbb{E}\left[\|\nabla F_i(w_{k,t}^i) - \frac{1}{n}\sum_{j=1}^{n}\nabla F_j(w_{k,t}^j)\|^2\right]. \tag{73}$$

Now, it remains to bound the last term in (73). Recall $\eta_i = \nabla F_i(w_{k,t}^i) - \nabla F_i(w_{k,t})$. First, note that, using $\|a + b\|^2 \leq 2\|a\|^2 + 2\|b\|^2$, we have

$$\|\nabla F_i(w_{k,t}^i) - \frac{1}{n}\sum_{j=1}^n \nabla F_j(w_{k,t}^j)\|^2 \leq 2\|\nabla F_i(w_{k,t}) - \frac{1}{n}\sum_{j=1}^n \nabla F_j(w_{k,t})\|^2 + 2\|\eta_i - \frac{1}{n}\sum_{j=1}^n \eta_j\|^2. \tag{74}$$

Substituting this bound in (73) and using Lemma 4.4 yields

$$\Sigma_{t+1} \leq (1 + \phi)\Sigma_t + 4(1 + \frac{1}{\phi})\beta^2(2\sigma_F^2 + \gamma_F^2) + 4(1 + \frac{1}{\phi})\beta^2 \frac{1}{n}\sum_{i=1}^n \mathbb{E}\left[\|\eta_i - \frac{1}{n}\sum_{j=1}^n \eta_j\|^2\right]. \tag{75}$$

Note that, using Cauchy-Schwarz inequality (71) with $a_1 = \eta_i, b_1 = 1$ and $a_l = 1/\sqrt{n}\eta_{l-1}, b_l = 1/\sqrt{n}$ for $l = 2, ..., n+1$, implies

$$\|\eta_i - \frac{1}{n}\sum_{j=1}^n \eta_j\|^2 \leq 2\left(\|\eta_i\|^2 + \frac{1}{n}\sum_{j=1}^n \|\eta_j\|^2\right)$$

$$\leq 2L_F^2\left(\|w_{k,t}^i - w_{k,t}\|^2 + \frac{1}{n}\sum_{j=1}^n \|w_{k,t}^i - w_{k,t}\|^2\right) \tag{76}$$

where the last inequality is obtained using Lemma 4.2 which states

$$\|\eta_i\| \leq L_F\|w_{k,t}^i - w_{k,t}\|. \tag{77}$$

Plugging (76) in (75) implies

$$\Sigma_{t+1} \leq \left(1 + \phi + 16(1 + \frac{1}{\phi})\beta^2 L_F^2\right)\Sigma_t + 4(1 + \frac{1}{\phi})\beta^2(2\sigma_F^2 + \gamma_F^2). \tag{78}$$

As a result, similar to (66), we obtain

$$\Sigma_{t+1} \leq 4\beta^2(1 + \frac{1}{\phi})(t + 1)\left(1 + \phi + 16(1 + \frac{1}{\phi})\beta^2 L_F^2\right)^t (2\sigma_F^2 + \gamma_F^2) \tag{79}$$

which gives us the desired result (57b).

Finally, to show (58), first note that for any $n$, we know

$$(1 + \frac{1}{n})^n \leq e. \tag{80}$$

Using this, along with the assumption $\beta \leq 1/(10L_F\tau)$ and the fact that $e^{0.2} \leq 2$, we immediately obtain (58a). To show the other one (58b), we use (57b) with $\phi = 1/(2\tau)$:

$$\phi + 16(1 + \frac{1}{\phi})\beta^2 L_F^2 = \frac{1}{2\tau} + 16(1 + 2\tau)\beta^2 L_F^2$$

$$\leq \frac{1}{2\tau} + 16(1 + 2\tau)\frac{1}{100\tau^2}$$

$$\leq \frac{1}{\tau} \tag{81}$$

where the first inequality follows from the assumption $\beta \leq 1/(10L_F\tau)$ and the last inequality is obtained using the trivial bound $1 + 2\tau \leq 3\tau$. Finally, using (81) along with (80) completes the proof. $\square$

## G Proof of Theorem 4.5

Although we only ask a fraction of agents to compute their local updates in Algorithm 1, here, and just for the sake of analysis, we assume all agents perform local updates. This is just for our analysis and we will not use all agents' updates in computing $w_{k+1}$. Also, from Proposition F.1, recall that $w_{k,t} = 1/n\sum_{i=1}^n w_{k,t}^i$.

Let $\mathcal{F}_{k+1}^t$ denote the $\sigma$-field generated by $\{w_{k+1,t}^i\}_{i=1}^n$. Note that, by Lemma 4.2, we know $F$ is smooth with gradient Lipschitz parameter $L_F$, and thus, by (17b), we have

$$F(\bar{w}_{k+1,t+1})$$

$$\leq F(\bar{w}_{k+1,t}) + \nabla F(\bar{w}_{k+1,t})^\top (\bar{w}_{k+1,t+1} - \bar{w}_{k+1,t}) + \frac{L_F}{2}\|\bar{w}_{k+1,t+1} - \bar{w}_{k+1,t}\|^2$$

$$\leq F(\bar{w}_{k+1,t}) - \beta \nabla F(\bar{w}_{k+1,t})^\top \left( \frac{1}{rn} \sum_{i \in \mathcal{A}_k} \tilde{\nabla} F_i(w_{k+1,t}^i) \right) + \frac{L_F}{2}\beta^2 \| \frac{1}{rn} \sum_{i \in \mathcal{A}_k} \tilde{\nabla} F_i(w_{k+1,t}^i)\|^2$$

$$(82)$$

where the last inequality is obtained using the fact that

$$\bar{w}_{k+1,t+1} = \frac{1}{rn} \sum_{i \in \mathcal{A}_k} w_{k+1,t+1}^i$$

$$= \frac{1}{rn} \sum_{i \in \mathcal{A}_k} \left( w_{k+1,t}^i - \beta \tilde{\nabla} F_i(w_{k+1,t}^i) \right)$$

$$= \bar{w}_{k+1,t} - \beta \frac{1}{rn} \sum_{i \in \mathcal{A}_k} \tilde{\nabla} F_i(w_{k+1,t}^i).$$

Taking expectation from both sides of (82) yields

$$\mathbb{E}\left[F(\bar{w}_{k+1,t+1})\right] \leq \mathbb{E}[F(\bar{w}_{k+1,t})] - \beta \mathbb{E}\left[ \nabla F(\bar{w}_{k+1,t})^\top \left( \frac{1}{rn} \sum_{i \in \mathcal{A}_k} \tilde{\nabla} F_i(w_{k+1,t}^i) \right) \right]$$

$$+ \frac{L_F}{2}\beta^2 \mathbb{E}\left[ \| \frac{1}{rn} \sum_{i \in \mathcal{A}_k} \tilde{\nabla} F_i(w_{k+1,t}^i)\|^2 \right] \qquad (83)$$

Next, note that

$$\frac{1}{rn} \sum_{i \in \mathcal{A}_k} \tilde{\nabla} F_i(w_{k+1,t}^i) = X + Y + Z + \frac{1}{rn} \sum_{i \in \mathcal{A}_k} \nabla F_i(\bar{w}_{k+1,t}) \qquad (84)$$

where

$$X = \frac{1}{rn} \sum_{i \in \mathcal{A}_k} \left( \tilde{\nabla} F_i(w_{k+1,t}^i) - \nabla F_i(w_{k+1,t}^i) \right), \qquad (85)$$

$$Y = \frac{1}{rn} \sum_{i \in \mathcal{A}_k} \left( \nabla F_i(w_{k+1,t}^i) - \nabla F_i(w_{k+1,t}) \right), \qquad (86)$$

$$Z = \frac{1}{rn} \sum_{i \in \mathcal{A}_k} \left( \nabla F_i(w_{k+1,t}) - \nabla F_i(\bar{w}_{k+1,t}) \right). \qquad (87)$$

We next bound the moments of $X$, $Y$, and $Z$, condition on $\mathcal{F}_{k+1}^t$. First, recall the Cauchy-Schwarz inequality

$$\left\| \sum_{i=1}^{rn} a_i b_i \right\|^2 \leq \left( \sum_{i=1}^{rn} \|a_i\|^2 \right) \left( \sum_{i=1}^{rn} \|b_i\|^2 \right). \qquad (88)$$

- Using this inequality with $a_i = (\tilde{\nabla} F_i(w_{k+1,t}^i) - \nabla F_i(w_{k+1,t}^i))/\sqrt{rn}$ and $b_l = 1/\sqrt{rn}$, we obtain

$$\|X\|^2 \leq \frac{1}{rn} \sum_{i \in \mathcal{A}_k} \left\| \tilde{\nabla} F_i(w_{k+1,t}^i) - \nabla F_i(w_{k+1,t}^i) \right\|^2, \qquad (89)$$

and hence, by using Lemma 4.3 along with the tower rule, we have

$$\mathbb{E}[\|X\|^2] = \mathbb{E}[\mathbb{E}[\|X\|^2 \mid \mathcal{F}_{k+1}^t]] \leq \sigma_F^2. \qquad (90)$$

- Regarding $Y$, note that by using Cauchy-Schwarz inequality (similar to what we did above) along with smoothness of $F_i$, we obtain

$$\|Y\|^2 \leq \frac{1}{rn} \sum_{i \in \mathcal{A}_k} \left\| \nabla F_i(w_{k+1,t}^i) - \nabla F_i(w_{k+1,t}) \right\|^2 \leq \frac{L_F^2}{rn} \sum_{i \in \mathcal{A}_k} \left\| w_{k+1,t}^i - w_{k+1,t} \right\|^2.$$
(91)

Again, taking expectation and using the fact that $\mathcal{A}_k$ is chosen uniformly at random, implies

$$\mathbb{E}[\|Y\|^2] = \mathbb{E}[\mathbb{E}[\|Y\|^2 \mid \mathcal{F}_{k+1}^t]]$$

$$\leq L_F^2 \mathbb{E}\left[ \mathbb{E}\left[ \frac{1}{rn} \sum_{i \in \mathcal{A}_k} \left\| w_{k+1,t}^i - w_{k+1,t} \right\|^2 \,\Big|\, \mathcal{F}_{k+1}^t \right] \right]$$

$$= L_F^2 \mathbb{E}\left[ \frac{1}{n} \sum_{i=1}^n \|w_{k,t}^i - w_{k,t}\|^2 \right]$$

$$\leq 35\beta^2 L_F^2 \tau(\tau - 1)(2\sigma_F^2 + \gamma_F^2)$$
(92)

where the last step follows from (58b) in Corollary F.2 along with the fact that $t \leq \tau - 1$.

- Regarding $Z$, first recall that if we have $n$ numbers $a_1, ..., a_n$ with mean $\mu = 1/n \sum_{i=1}^n a_i$ and variance $\sigma^2 = 1/n \sum_{i=1}^n |a_i - \mu|^2$, and we take a subset of them $\{a_i\}_{i \in \mathcal{A}}$ with size $|\mathcal{A}| = rn$ by sampling without replacement, then we have

$$\mathbb{E}\left[ \left| \frac{\sum_{i \in \mathcal{A}} a_i}{rn} - \mu \right|^2 \right] = \frac{\sigma^2}{rn} \left( 1 - \frac{rn - 1}{n - 1} \right) = \frac{\sigma^2(1 - r)}{r(n - 1)}.$$
(93)

Using this, we have

$$\mathbb{E}\left[ \|\bar{w}_{k+1,t} - w_{k+1,t}\|^2 \mid \mathcal{F}_{k+1}^t \right] \leq \frac{(1 - r)/n \sum_{i=1}^n \|w_{k+1,t}^i - w_{k+1,t}\|^2}{r(n - 1)},$$
(94)

and hence, by taking expectation from both sides and using the tower rule along with (58b) in Corollary F.2, we obtain

$$\mathbb{E}\left[ \|\bar{w}_{k+1,t} - w_{k+1,t}\|^2 \right] \leq \frac{35(1 - r)\beta^2 \tau(\tau - 1)(2\sigma_F^2 + \gamma_F^2)}{r(n - 1)}.$$
(95)

Next, note that by using Cauchy-Schwarz inequality (88), with $a_i = (\nabla F_i(w_{k+1,t}) - \nabla F_i(\bar{w}_{k+1,t}))/\sqrt{rn}$ and $b_i = 1/\sqrt{rn}$, we have

$$\|Z\|^2 \leq \frac{1}{rn} \sum_{i \in \mathcal{A}_k} \left\| \nabla F_i(w_{k+1,t}) - \nabla F_i(\bar{w}_{k+1,t}) \right\|^2$$

$$\leq \frac{L_F^2}{rn} \sum_{i \in \mathcal{A}_k} \|w_{k+1,t} - \bar{w}_{k+1,t}\|^2 = L_F^2 \|\bar{w}_{k+1,t} - w_{k+1,t}\|^2$$
(96)

where the last inequality is obtained using smoothness of $F_i$ (Lemma 4.2). Now, taking expectation from both sides and using (95) yields

$$\mathbb{E}[\|Z\|^2] \leq \frac{35(1 - r)\beta^2 L_F^2 \tau(\tau - 1)(2\sigma_F^2 + \gamma_F^2)}{r(n - 1)}.$$
(97)

Now, getting back to (83), we first lower bound the term

$$\mathbb{E}\left[ \nabla F(\bar{w}_{k+1,t})^\top \left( \frac{1}{rn} \sum_{i \in \mathcal{A}_k} \tilde{\nabla} F_i(w_{k+1,t}^i) \right) \right].$$

To do so, note that, by (84), we have

$$\mathbb{E}\left[\nabla F(\bar{w}_{k+1,t})^{\top}\left(\frac{1}{rn}\sum_{i\in\mathcal{A}_k}\tilde{\nabla}F_i(w_{k+1,t}^i)\right)\right]$$

$$= \mathbb{E}\left[\nabla F(\bar{w}_{k+1,t})^{\top}\left(X+Y+Z+\frac{1}{rn}\sum_{i\in\mathcal{A}_k}\nabla F_i(\bar{w}_{k+1,t})\right)\right]$$

$$\geq \mathbb{E}\left[\nabla F(\bar{w}_{k+1,t})^{\top}\left(\frac{1}{rn}\sum_{i\in\mathcal{A}_k}\nabla F_i(\bar{w}_{k+1,t})\right)\right] - \left\|\mathbb{E}\left[\nabla F(\bar{w}_{k+1,t})^{\top}X]\right]\right\|$$

$$- \frac{1}{4}\mathbb{E}[\|\nabla F(\bar{w}_{k+1,t})\|^2] - \mathbb{E}[\|Y+Z\|^2] \qquad (98)$$

where the last inequality is obtained using the fact that

$$\mathbb{E}\left[\nabla F(\bar{w}_{k+1,t})^{\top}(Y+Z)\right] \leq \frac{1}{4}\mathbb{E}[\|\nabla F(\bar{w}_{k+1,t})\|^2] + \mathbb{E}[\|Y+Z\|^2].$$

Now, we bound terms in (98) separately. First, note that by tower rule we have

$$\mathbb{E}\left[\nabla F(\bar{w}_{k+1,t})^{\top}\left(\frac{1}{rn}\sum_{i\in\mathcal{A}_k}\nabla F_i(\bar{w}_{k+1,t})\right)\right]$$

$$= \mathbb{E}\left[\mathbb{E}\left[\nabla F(\bar{w}_{k+1,t})^{\top}\left(\frac{1}{rn}\sum_{i\in\mathcal{A}_k}\nabla F_i(\bar{w}_{k+1,t})\right)\,\Big|\,\mathcal{F}_{k+1}^t\right]\right]$$

$$= \mathbb{E}\left[\nabla F(\bar{w}_{k+1,t})^{\top}\mathbb{E}\left[\left(\frac{1}{rn}\sum_{i\in\mathcal{A}_k}\nabla F_i(\bar{w}_{k+1,t})\right)\,\Big|\,\mathcal{F}_{k+1}^t\right]\right]$$

$$= \mathbb{E}\left[\|\nabla F(\bar{w}_{k+1,t})\|^2\right] \qquad (99)$$

where the last equality is obtained using the fact that $\mathcal{A}_k$ is chosen uniformly at random, and thus,

$$\mathbb{E}\left[\left(\frac{1}{rn}\sum_{i\in\mathcal{A}_k}\nabla F_i(\bar{w}_{k+1,t})\right)\,\Big|\,\mathcal{F}_{k+1}^t\right] = \frac{1}{n}\sum_{i=1}^{n}\nabla F_i(\bar{w}_{k+1,t}).$$

Second, note that

$$\mathbb{E}\left[\nabla F(\bar{w}_{k+1,t})^{\top}X\right] = \mathbb{E}\left[\mathbb{E}\left[\nabla F(\bar{w}_{k+1,t})^{\top}X\,\Big|\,\mathcal{F}_{k+1}^t\right]\right]$$

$$= \mathbb{E}\left[\nabla F(\bar{w}_{k+1,t})^{\top}\mathbb{E}\left[X\,\Big|\,\mathcal{F}_{k+1}^t\right]\right].$$

As a result, we have

$$\left\|\mathbb{E}\left[\nabla F(\bar{w}_{k+1,t})^{\top}X]\right]\right\| = \left\|\mathbb{E}\left[\nabla F(\bar{w}_{k+1,t})^{\top}\mathbb{E}\left[X\,\Big|\,\mathcal{F}_{k+1}^t\right]\right]\right\|$$

$$\leq \frac{1}{4}\mathbb{E}[\|\nabla F(\bar{w}_{k+1,t})\|^2] + \mathbb{E}\left[\left\|\mathbb{E}\left[X\,\Big|\,\mathcal{F}_{k+1}^t\right]\right\|^2\right]$$

$$\leq \frac{1}{4}\mathbb{E}[\|\nabla F(\bar{w}_{k+1,t})\|^2] + \frac{4\alpha^2 L^2\sigma_G^2}{D} \qquad (100)$$

where the last inequality follows from Lemma 4.3. Third, note that by Cauchy-Schwarz inequality,

$$\mathbb{E}[\|Y+Z\|^2] \leq 2\left(\mathbb{E}[\|Y\|^2] + \mathbb{E}[\|Z\|^2]\right)$$

$$\leq 70\beta^2 L_F^2\tau(\tau-1)(2\sigma_F^2+\gamma_F^2)\left(1+\frac{1-r}{r(n-1)}\right)$$

$$\leq 140\beta^2 L_F^2\tau(\tau-1)(2\sigma_F^2+\gamma_F^2) \qquad (101)$$

where second inequality is obtained using (92) and (97). Plugging (99), (100), and (101) in (98) implies

$$\mathbb{E}\left[\nabla F(\bar{w}_{k+1,t})^{\top}\left(\frac{1}{rn}\sum_{i\in\mathcal{A}_k}\tilde{\nabla}F_i(w_{k+1,t}^i)\right)\right]$$

$$\geq \frac{1}{2}\mathbb{E}[\|\nabla F(\bar{w}_{k+1,t})\|^2] - 140\beta^2 L_F^2 \tau(\tau-1)(2\sigma_F^2 + \gamma_F^2) - \frac{4\alpha^2 L^2 \sigma_G^2}{D}. \qquad (102)$$

Next, we characterize an upper bound for the other term in (83):

$$\mathbb{E}\left[\|\frac{1}{rn}\sum_{i\in\mathcal{A}_k}\tilde{\nabla}F_i(w_{k+1,t}^i)\|^2\right]$$

Note that, by (84) we have

$$\|\frac{1}{rn}\sum_{i\in\mathcal{A}_k}\tilde{\nabla}F_i(w_{k+1,t}^i)\|^2 \leq 2\|X+Y+Z\|^2 + 2\|\frac{1}{rn}\sum_{i\in\mathcal{A}_k}\nabla F_i(\bar{w}_{k+1,t})\|^2, \qquad (103)$$

and thus, by (101) along with (90), we have

$$\mathbb{E}\left[\|\frac{1}{rn}\sum_{i\in\mathcal{A}_k}\tilde{\nabla}F_i(w_{k+1,t}^i)\|^2\right]$$

$$\leq 2\mathbb{E}\left[\|\frac{1}{rn}\sum_{i\in\mathcal{A}_k}\nabla F_i(\bar{w}_{k+1,t})\|^2\right] + 4\sigma_F^2 + 560\beta^2 L_F^2 \tau(\tau-1)(2\sigma_F^2 + \gamma_F^2). \qquad (104)$$

Note that, $\mathbb{E}\left[1/(rn)\sum_{i\in\mathcal{A}_k}\nabla F_i(\bar{w}_{k+1,t}) \mid \mathcal{F}_{k+1}^t\right] = \nabla F(\bar{w}_{k+1,t})$, since $\mathcal{A}_k$ is chosen uniformly at random. Also, by Lemma 4.4, we have

$$\frac{1}{n}\mathbb{E}\left[\|\nabla F_i(\bar{w}_{k+1,t}) - \nabla F(\bar{w}_{k+1,t})\|^2 \,\Big|\, \mathcal{F}_{k+1}^t\right] \leq \gamma_F^2,$$

and thus, by (93), we have

$$\mathbb{E}\left[\|\frac{1}{rn}\sum_{i\in\mathcal{A}_k}\nabla F_i(\bar{w}_{k+1,t})\|^2\right] \leq \mathbb{E}\left[\|\nabla F(\bar{w}_{k+1,t})\|^2\right] + \frac{\gamma_F^2(1-r)}{r(n-1)}. \qquad (105)$$

Plugging (105) in (104), we obtain

$$\mathbb{E}\left[\left\|\frac{1}{rn}\sum_{i\in\mathcal{A}_k}\tilde{\nabla}F_i(w_{k+1,t}^i)\right\|^2\right]$$

$$\leq 2\mathbb{E}\left[\|\nabla F(\bar{w}_{k+1,t})\|^2\right] + \frac{2\gamma_F^2(1-r)}{r(n-1)} + 4\sigma_F^2 + 560\beta^2 L_F^2 \tau(\tau-1)(2\sigma_F^2 + \gamma_F^2). \qquad (106)$$

Substituting (106) and (102) in (83) implies

$$\mathbb{E}\left[F(\bar{w}_{k+1,t+1})\right]$$

$$\leq \mathbb{E}[F(\bar{w}_{k+1,t})] - \beta(1/2 - \beta L_F)\mathbb{E}\left[\|\nabla F(\bar{w}_{k+1,t})\|^2\right]$$

$$+ 140(1+2\beta L_F)\beta^3 L_F^2 \tau(\tau-1)(2\sigma_F^2 + \gamma_F^2) + \beta^2 L_F\left(2\sigma_F^2 + \frac{\gamma_F^2(1-r)}{r(n-1)}\right) + \frac{4\beta\alpha^2 L^2 \sigma_G^2}{D}$$

$$\leq \mathbb{E}[F(\bar{w}_{k+1,t})] - \frac{\beta}{4}\mathbb{E}\left[\|\nabla F(\bar{w}_{k+1,t})\|^2\right] + \beta\sigma_T^2. \qquad (107)$$

where

$$\sigma_T^2 := 280(\beta L_F)^2 \tau(\tau-1)(2\sigma_F^2 + \gamma_F^2) + \beta L_F\left(2\sigma_F^2 + \frac{\gamma_F^2(1-r)}{r(n-1)}\right) + \frac{4\alpha^2 L^2 \sigma_G^2}{D} \qquad (108)$$

the last inequality is obtained using $\beta \leq 1/(10\tau L_F)$. Summing up (107) for all $t = 0, ..., \tau-1$, we obtain

$$\mathbb{E}\left[F(w_{k+1})\right] \leq \mathbb{E}\left[F(w_k)\right] - \frac{\beta\tau}{4}\left(\frac{1}{\tau}\sum_{t=0}^{\tau-1}E\left[\|\nabla F(\bar{w}_{k+1,t})\|^2\right]\right) + \beta\tau\sigma_T^2 \qquad (109)$$

where we used the fact that $\bar{w}_{k+1,\tau} = w_{k+1}$. Finally, summing up (109) for $k = 0, ..., K-1$ implies

$$\mathbb{E}\left[F(w_K)\right] \leq F(w_0) - \frac{\beta\tau K}{4}\left(\frac{1}{\tau K}\sum_{k=0}^{K-1}\sum_{t=0}^{\tau-1}E\left[\|\nabla F(\bar{w}_{k+1,t})\|^2\right]\right) + \beta\tau K\sigma_T^2. \tag{110}$$

As a result, we have

$$\frac{1}{\tau K}\sum_{k=0}^{K-1}\sum_{t=0}^{\tau-1}E\left[\|\nabla F(\bar{w}_{k+1,t})\|^2\right] \leq \frac{4}{\beta\tau K}\left(F(w_0) - \mathbb{E}\left[F(w_K)\right] + \beta\tau K\sigma_T^2\right)$$

$$\leq \frac{4(F(w_0) - F^*)}{\beta\tau K} + 4\sigma_T^2 \tag{111}$$

which gives us the desired result.

**Remark G.1.** *As stated in Remark 4.8, we could easily extend our analysis to the case with diminishing stepsize. In particular, by using $\beta_k$ as the stepsize at iteration $k$, the descent result (109) holds with $\beta = \beta_k$. Hence, summing up this equation for $k = 0, ..., K-1$, we recover the same complexity bounds using $\beta_k = \mathcal{O}(1/\sqrt{\tau k})$.*

## H  On First-Order Approximations of Per-FedAvg

As we stated previously, the Per-FedAvg method, same as MAML, requires computing Hessian-vector product which is computationally costly in some applications. As a result, one may consider using the first-order approximation of the update rule for the Per-FedAvg algorithm. The main goal of this section is to show how our analysis can be extended to the case that we either drop the second-order term or approximate the Hessian-vector product using first-order techniques.

To do so, we show that it suffices to only extend the result in Lemma 4.3 for the first-order approximation settings and find $\tilde{\sigma}_F$ such that

$$\left\|\mathbb{E}\left[\tilde{\nabla}F_i(w) - \nabla F_i(w)\right]\right\| \leq m_F,$$

$$\mathbb{E}\left[\left\|\tilde{\nabla}F_i(w) - \nabla F_i(w)\right\|^2\right] \leq \tilde{\sigma}_F^2.$$

One can easily check that the rest of analysis does not change, and the final result (Theorem 4.5) holds if we just replace $\sigma_F$ by $\tilde{\sigma}_F$ and $\alpha^2 L^2 \sigma_G^2/D$ by $m_F^2$.

We next focus on two different approaches, developed for MAML formulation, for approximating the Hessian-vector product, and show how we can characterize $m_F$ and $\tilde{\sigma}_F$ for both cases:

• **Ignoring the second-order term:** Authors in [2] suggested to simply ignore the second-order term in the update of MAML to reduce the computation cost of MAML, i.e., to replace $\tilde{\nabla}F_i(w)$ with

$$\tilde{\nabla}f_i\left(w - \alpha\tilde{\nabla}f_i(w, \mathcal{D}), \mathcal{D}'\right). \tag{112}$$

This approach is known as First-Order MAML (FO-MAML), and it has been shown that it performs relatively well in many cases [2]. In particular, [31] characterized the convergence properties of FO-MAML for the centralized MAML problem. Next, we characterize the mean and variance of this gradient approximation.

**Lemma H.1.** *Assume that we estimate $\nabla F_i(w)$ by (112) where $\mathcal{D}$ and $\mathcal{D}'$ are independent batches with size $D$ and $D'$, respectively. Suppose that the conditions in Assumptions 2-4 are satisfied. Then, for any $\alpha \in [0, 1/L]$ and $w \in \mathbb{R}^d$, we have*

$$\left\|\mathbb{E}\left[\tilde{\nabla}F_i(w) - \nabla F_i(w)\right]\right\| \leq m_F^{FO} := \alpha L\left(\frac{\sigma_G}{\sqrt{D}} + B\right),$$

$$\mathbb{E}\left[\left\|\tilde{\nabla}F_i(w) - \nabla F_i(w)\right\|^2\right] \leq (\tilde{\sigma}_F^{FO})^2 := 2\sigma_G^2\left(\frac{1}{D'} + \frac{(\alpha L)^2}{D}\right) + 2(\alpha LB)^2.$$

*Proof.* In fact, in this case, $\tilde{\nabla}F_i(w)$ is approximating

$$G_i(w) := \nabla f_i\left(w - \alpha\nabla f_i(w)\right). \tag{113}$$

To bound $m_F^{FO}$, note that

$$\left\|\mathbb{E}\left[\tilde{\nabla}F_i(w) - \nabla F_i(w)\right]\right\| \leq \left\|\mathbb{E}\left[\tilde{\nabla}F_i(w) - G_i(w)\right]\right\| + \left\|\mathbb{E}\left[G_i(w) - \nabla F_i(w)\right]\right\| \tag{114}$$

$$\leq \frac{\alpha L \sigma_G}{\sqrt{D}} + \alpha L B \tag{115}$$

where the first term follows from (33) in the proof of Lemma 4.3 in Appendix D, and the second term is obtained using

$$\|G_i(w) - \nabla F_i(w)\| = \alpha \left\|\nabla^2 f_i(w) \nabla f_i\left(w - \alpha \nabla f_i(w)\right)\right\|$$

$$\leq \alpha \|\nabla^2 f_i(w)\| \cdot \|\nabla f_i\left(w - \alpha \nabla f_i(w)\right)\| \leq \alpha L B \tag{116}$$

where the first inequality follows from the matrix norm definition and the last inequality is obtained using Assumption 2.

To characterize $\tilde{\sigma}_F^{FO}$, note that

$$\mathbb{E}\left[\left\|\tilde{\nabla}F_i(w) - \nabla F_i(w)\right\|^2\right] \leq 2\mathbb{E}\left[\left\|\tilde{\nabla}F_i(w) - G_i(w)\right\|^2\right] + 2\mathbb{E}\left[\|G_i(w) - \nabla F_i(w)\|^2\right]. \tag{117}$$

We bound these two terms separately. Note that we have already bounded the first term in Appendix D (see (36)), and we have

$$\mathbb{E}\left[\left\|\tilde{\nabla}F_i(w) - G_i(w)\right\|^2\right] \leq \sigma_G^2 \left(\frac{1}{D'} + \frac{(\alpha L)^2}{D}\right). \tag{118}$$

Plugging (118) and (116) into (117), we obtain the desired result. $\qquad\square$

Note that while the first term in $\tilde{\sigma}_F^{FO}$ can be made arbitrary small by choosing $D$ and $D'$ large enough, this is not the case for the second term. However, the second term is also negligible if $\alpha$ is small enough. Yet this bound suggests that this approximation introduces a non-vanishing error term which is directly carried to the final result (Theorem 4.5).

● **Estimating Hessian-vector product using gradient differences:** In the context of MAML problem, it has been shown that the update of FO-MAML leads to an additive error that does not vanish as time progresses. To resolve this matter, [31] introduced another variant of MAML, called HF-MAML, which approximates the Hessian-vector product by gradient differences. More formally, the idea behind their method is that for any function $g$, the product of the Hessian $\nabla^2 g(w)$ by any vector $v$ can be approximated by

$$\frac{\nabla g(w + \delta v) - \nabla g(w - \delta v)}{2\delta} \tag{119}$$

with an error of at most $\rho\delta\|v\|^2$, where $\rho$ is the parameter for Lipschitz continuity of the Hessian of $g$. Building on this idea, in Per-FedAvg update rule, we can replace $\tilde{\nabla}F_i(w)$ by

$$\tilde{\nabla}f_i\left(w - \alpha\tilde{\nabla}f_i(w, \mathcal{D}), \mathcal{D}'\right) - \alpha\tilde{d}_i(w) \tag{120}$$

where

$$\tilde{d}_i(w) := \frac{\tilde{\nabla}f_i\left(w + \delta\tilde{\nabla}f_i(w - \alpha\tilde{\nabla}f_i(w, \mathcal{D}), \mathcal{D}'), \mathcal{D}''\right) - \tilde{\nabla}f_i\left(w - \delta\tilde{\nabla}f_i(w - \alpha\tilde{\nabla}f_i(w, \mathcal{D}), \mathcal{D}'), \mathcal{D}''\right)}{2\delta}. \tag{121}$$

For this approximation, we have the following result, which shows that we have an additional degree of freedom ($\delta$) to control the error term that does not decreased with increasing batch sizes.

**Lemma H.2.** *Assume that we estimate $\nabla F_i(w)$ by (120) where $\mathcal{D}$, $\mathcal{D}'$, and $D''$ are independent batches with size $D$, $D'$, and $D''$, respectively. Suppose that the conditions in Assumptions 2-4 are satisfied. Then, for any $\alpha \in [0, 1/L]$ and $w \in \mathbb{R}^d$, we have*

$$\left\|\mathbb{E}\left[\tilde{\nabla}F_i(w) - \nabla F_i(w)\right]\right\| \leq m_F^{HF} := \alpha\left(\frac{2L\sigma_G}{\sqrt{D}} + \frac{L\sigma_G}{\sqrt{D'}} + \rho\delta B^2\right),$$

$$\mathbb{E}\left[\left\|\tilde{\nabla}F_i(w) - \nabla F_i(w)\right\|^2\right] \leq (\tilde{\sigma}_F^{HF})^2 := 6\sigma_G^2\left(\frac{2(\alpha L)^2}{D} + \frac{2}{D'} + \frac{\alpha^2}{2\delta^2 D''}\right) + 2(\alpha\rho\delta)^2 B^4.$$

*Proof.* Note that, this time $\tilde{\nabla}F_i(w)$ is approximating

$$G_i^{'}(w) := \nabla f_i\left(w - \alpha\nabla f_i(w)\right) - \alpha d_i(w) \tag{122}$$

where

$$d_i(w) := \frac{\nabla f_i\left(w + \delta\nabla f_i\left(w - \alpha\nabla f_i(w)\right)\right) - \nabla f_i\left(w - \delta\nabla f_i\left(w - \alpha\nabla f_i(w)\right)\right)}{2\delta} \tag{123}$$

is the term approximating $\nabla^2 f_i(w)\nabla f_i\left(w - \alpha\nabla f_i(w)\right)$. Below, we characterize $\tilde{\sigma}_F^{HF}$, and $m_F^{HF}$ can be done similarly.

Similar to (117), we have

$$\mathbb{E}\left[\left\|\tilde{\nabla}F_i(w) - \nabla F_i(w)\right\|^2\right] \le 2\mathbb{E}\left[\left\|\tilde{\nabla}F_i(w) - G_i^{'}(w)\right\|^2\right] + 2\mathbb{E}\left[\left\|G_i^{'}(w) - \nabla F_i(w)\right\|^2\right]. \tag{124}$$

We again bound both terms separately. To simplify the notation, let us define

$$g_i(w) := \nabla f_i\left(w - \alpha\nabla f_i(w)\right), \quad \tilde{g}_i(w) := \tilde{\nabla}f_i\left(w - \alpha\tilde{\nabla}f_i(w,\mathcal{D}),\mathcal{D}'\right). \tag{125}$$

First, note that, using $(a + b + c)^2 \le 3(a^2 + b^2 + c^2)$ for $a, b, c \ge 0$, we have

$$\left\|\tilde{\nabla}F_i(w) - G_i^{'}(w)\right\|^2$$

$$\le 3\|\tilde{g}_i(w) - g_i(w)\|^2 + \frac{3\alpha^2}{4\delta^2}\left\|\tilde{\nabla}f_i\left(w + \delta\tilde{g}_i(w),\mathcal{D}''\right) - \nabla f_i(w + \delta g_i(w))\right\|^2$$

$$+ \frac{3\alpha^2}{4\delta^2}\left\|\tilde{\nabla}f_i\left(w - \delta\tilde{g}_i(w),\mathcal{D}''\right) - \nabla f_i(w - \delta g_i(w))\right\|^2. \tag{126}$$

Taking expectation from both sides, along with using (118), we have

$$\mathbb{E}\left[\left\|\tilde{\nabla}F_i(w) - G_i^{'}(w)\right\|^2\right]$$

$$\le 3\sigma_G^2\left(\frac{1}{D'} + \frac{(\alpha L)^2}{D}\right) + \frac{3\alpha^2}{4\delta^2}\left(\mathbb{E}\left[\left\|\tilde{\nabla}f_i\left(w + \delta\tilde{g}_i(w),\mathcal{D}''\right) - \nabla f_i(w + \delta g_i(w))\right\|^2\right]\right.$$

$$+ \mathbb{E}\left[\left\|\tilde{\nabla}f_i\left(w - \delta\tilde{g}_i(w),\mathcal{D}''\right) - \nabla f_i(w - \delta g_i(w))\right\|^2\right]\right)$$

$$\le 3\sigma_G^2\left(\frac{\alpha^2}{2\delta^2 D''} + \frac{1}{D'} + \frac{(\alpha L)^2}{D}\right) + \frac{3\alpha^2}{4\delta^2}\left(\mathbb{E}\left[\|\nabla f_i\left(w + \delta\tilde{g}_i(w)\right) - \nabla f_i(w + \delta g_i(w))\|^2\right]\right.$$

$$+ \mathbb{E}\left[\|\nabla f_i\left(w - \delta\tilde{g}_i(w)\right) - \nabla f_i(w - \delta g_i(w))\|^2\right]\right) \tag{127}$$

where (127) is obtained using the fact that $\mathcal{D}''$ is independent from $\mathcal{D}$ and $\mathcal{D}'$ which implies

$$\mathbb{E}\left[\left\|\tilde{\nabla}f_i\left(w \pm \delta\tilde{g}_i(w),\mathcal{D}''\right) - \nabla f_i(w \pm \delta g_i(w))\right\|^2\right] \le \frac{\sigma_G^2}{D''}$$

$$+ \mathbb{E}\left[\|\nabla f_i\left(w \pm \delta\tilde{g}_i(w)\right) - \nabla f_i(w \pm \delta g_i(w))\|^2\right].$$

Next, note that Assumption 2 yields

$$\|\nabla f_i\left(w \pm \delta\tilde{g}_i(w)\right) - \nabla f_i(w \pm \delta g_i(w))\| \le \delta L\|\tilde{g}_i(w) - g_i(w)\|.$$

Plugging this bound into (127) and using (117) implies

$$\mathbb{E}\left[\left\|\tilde{\nabla}F_i(w) - G_i^{'}(w)\right\|^2\right] \le 3\sigma_G^2\left(\frac{\alpha^2}{2\delta^2 D''} + (1 + \frac{(\alpha L)^2}{2})\left(\frac{1}{D'} + \frac{(\alpha L)^2}{D}\right)\right)$$

$$\le 3\sigma_G^2\left(\frac{2(\alpha L)^2}{D} + \frac{2}{D'} + \frac{\alpha^2}{2\delta^2 D''}\right) \tag{128}$$

where the last inequality is obtained using $\alpha L \le 1$.

Bounding the second term in (124) is more straightforward as we have

$$\left\|G_i^{'}(w) - \nabla F_i(w)\right\| = \alpha\left\|d_i(w) - \nabla^2 f_i(w)\nabla f_i\left(w - \alpha\nabla f_i(w)\right)\right\| \le \alpha\rho\delta\|g_i(w)\|^2 \le \alpha\rho\delta B^2. \tag{129}$$

Plugging (128) and (129) into (124) gives us the desired result. $\qquad\square$

Table 2: Illustration of the our experiment's setting
Image Classes

| | 1 | 2 | $\cdots$ | 5 | 6 | 7 | $\cdots$ | 10 |
|---|---|---|---|---|---|---|---|---|
| **1** | a | a | $\cdots$ | a | 0 | 0 | $\cdots$ | 0 |
| **2** | a | a | $\cdots$ | a | 0 | 0 | $\cdots$ | 0 |
| $\vdots$ | $\vdots$ | $\vdots$ | $\ddots$ | $\vdots$ | $\vdots$ | $\vdots$ | $\ddots$ | $\vdots$ |
| **5** | a | a | $\cdots$ | a | 0 | 0 | $\cdots$ | 0 |
| **6** | a/2 | 0 | $\cdots$ | 0 | 2a | 0 | $\cdots$ | 0 |
| **7** | 0 | a/2 | $\cdots$ | 0 | 0 | 2a | $\cdots$ | 0 |
| $\vdots$ | $\vdots$ | $\vdots$ | $\ddots$ | $\vdots$ | $\vdots$ | $\vdots$ | $\ddots$ | $\vdots$ |
| **10** | 0 | 0 | $\cdots$ | a/2 | 0 | 0 | $\cdots$ | 2a |

Groups of Users

a: Comparison in terms of runtime

b: Comparison in terms of number of iterations

# I   More on Numerical Experiments

In this section, we discuss our further results on numerical experiments. We thank the anonymous reviewers for their suggestions on adding this results, and we are looking forward to further explore our method from numerical point of view in future works.

First, in Table 2. we provide an illustration of the numerical setting in Section 5.

Second, in Figure 1a, we illustrate the average test accuracy of all studied algorithms with respect to time. As this figure shows, Per-FedAvg (HF) achieves higher level of accuracy compared to the regular Fed-Avg with local updates within the same computation time.

Third, we also compare our method with ARUBA [36]. To do so, we also report the output of FedAvg+ARUBA after refinement for each user. In particular, we consider $\tau = 4$ and $K = 1000$, and also tune hyper-parameters of ARUBA for a fair comparison. The final accuracy of all algorithms is as follows: Per-FedAvg(FO): $34.04 \pm 0.08$, Fed-Avg+ARUBA (with refinement): $36.74 \pm 0.1$, Per-FedAvg(HF): $43.73 \pm 0.11$. In Figure 1b, we have also depicted *one realization* of training path, just to provide intuition on the convergence speed of these methods.