[Reviews · NeurIPS 2020]

Review 1

Summary and Contributions: Post-rebuttal&discussion: thank your for the response, I'm sticking to my score (7). ***** In this paper, the authors analyze a federated learning method based on the minimization of the average of the agents' losses perturbed by a local gradient step. This perturbation in the individual losses is based on the MAML framework (from [2] and subsequently studied) and aims at providing a global models that performs well locally after being "personalized" by a gradient step.

Strengths: The joint learning of a personal model and a global meta-model is an interesting and timely topic, especially for federated learning applications. The authors analyze the optimization process stemming from the usual federated optimization rationale (agent subsampling, several stochastic gradient steps per agent iteration). They clearly motivate their assumptions and connect their result with usual federated averaging.

Weaknesses: - Concerning the main result (Th 4.5): *[let sigma_G=0] by taking appropriately tau and K, you can choose beta small enough to get epsilon accuracy. Would it be possible following you analysis to use a decreasing stepsize strategy? *[concerning the term alpha^2 sigma_G^2/D] to make the accuracy decrease to a thought value (without touching alpha), one has to reduce the variance of gradients (either by taking bigger batches or maybe variance reduction). A natural question is then: how does you results compare with the ones of [37] in the deterministic case? (In other words, what does strong convexity bring and what does non-convexity take?). - For the numerical experiments: * you do "one step of stochastic gradient" after FedAvg and then evaluate. Given the formulation of (3), maybe one pass on the data or one full gradient would maybe be more reasonable. * Why not implement Per-FedAvg (at least for comparison with the FO and HF approximations on a smaller example)?

Correctness: For the numerical experiments, you run all methods for K=1000 rounds, but the number of oracle calls (and data points sampled) is different for all 3 algorithms so it may be interesting to take that into account to have a better view of what the actual wallclock performance would look like.

Clarity: The paper is nicely written, providing a clear mathematical formulation of the problem and the method. The assumptions are especially well discussed. - the wording of lines 266-268 is a bit odd.

Relation to Prior Work: The relation with prior work is clear except on two points: * you adopt the personalization scheme of MAML but I feel that the interest of problem (3) compared to either a. the proximal average of the f_i or b. the limit point of local GD is not sufficiently discussed. * the mathematical and practical differences with [37] should be made clearer.

Reproducibility: Yes

Additional Feedback:


Review 2

Summary and Contributions: This paper considers the problem of federated meta-learning. They propose an algorithm based on the federated averaging algorithm and analyze its convergence behavior.

Strengths: Theoretical grounding: did not check the proof thoroughly, but it seems to be ok Empirical Evaluation: not quite adequate. significance and novelty: not quite novel relevance: relevant See additional feedback

Weaknesses: See additional feedback

Correctness: Did not check the proofs thoroughly The empirical section is very limited and not convincing

Clarity: mostly clear

Relation to Prior Work: Not quite (see additional feedback)

Reproducibility: Yes

Additional Feedback: This work considers the problem of federated meta-learning. For this setup the authors propose an algorithm based on the federated averaging and analyze its convergence behavior. My main concern with the paper is lack of novelty. As the authors have mentioned, the idea of using meta learning for personalization in federated learning is not novel. In fact, the analysis part is also not quite novel; similar lines of work are in [32, 33, 38]. My other concern with this paper is that it does not provide enough comparison, empirically and theoretically, with the previous works, specifically works such as [32, 38]. In fact, the empirical results are so limited that drawing any conclusions from them is impossible. For example, the experiments do not compare with many algorithms, and do not show convergence speed. It is not even quite clear if the hyper-params for different algorithms are tuned. Or how would the algorithm behave with respect to different levels of data heterogeneity. In Corollary 4.6. the conditions are not quite correct. Based on Definition 4.1. \epsilon-stationarity happens when E(\|\nabla f(w)\|)<=\epsilon. While these conditions only provide evidence for E(\|\nabla f(w)\|^2)<=\epsilon Lines 207-208: [42] only provides sample complexity for Wasserstein distance in 1-dimensional case. In general, the convergence speed of Wasserstein distance in d dimension is exponentially slow in d; e.g. see [*]. [*] Weed and Bach, “Sharp asymptotic and finite-sample rates of convergence of empirical measures in Wasserstein distance” --- As I mentioned in my review, my main concern with the paper is novlety and lack of meaningful contribution. I am not convinced even after reading author's rebuttal. The formulations are not new and the analysis is not particularly new either. The authors mention the challenges in analysis due to multiple local steps. But my issue with this challenge is that from the paper it is not even clear such a local step is actually useful or not. The analysis does not show the benefits and the experiments are so thin that it is hard to arrive at any conclusions based on them.


Review 3

Summary and Contributions: This paper combines federated learning with model-agnostic meta-learning, proposing a method namely Per-FedAvg. The proposed method can handle special yet realistic cases of federated learning where different users require personalized models for downstream tasks. Post rebuttal: most of my concerns are addressed yet the author should carefully improve in their final version. I would like to stick to my pre-rebuttal score 6.

Strengths: 1) The proposed method is simple and effective for the FL+MAML task. 2) Based on my evaluation, the theoretical analysis seems sound.

Weaknesses: 1) The authors talk about an NLP example for the application of FL+MAML. However, only computer vision tasks are considered in experiments, which creates inconsistency in this paper. 2) The clarity of this paper needs to be improved, especially on the details. 3) Not enough details are provided to reproduce the reported result.

Correctness: The theoretical claims and method seem sound. And the empirical methodology is correct.

Clarity: 1) I have trouble understanding the experimental setup in L266-268. Let's say, for MNIST, here is my guess of an example setup: digit 0 1 2 3 4 5 6 7 8 9 user0 196 0 0 0 0 0 0 0 0 0 user1 0 196 0 0 0 0 0 0 0 0 user2 0 0 196 0 0 0 0 0 0 0 user3 0 0 0 196 0 0 0 0 0 0 user4 0 0 0 0 196 0 0 0 0 0 user5 98 0 0 0 0 392 0 0 0 0 user6 0 98 0 0 0 0 392 0 0 0 user7 0 0 98 0 0 0 0 392 0 0 user8 0 0 0 98 0 0 0 0 392 0 user9 0 0 0 0 98 0 0 0 0 392 Is it correct? Does the image set overlap between user0 & user5? I think a more detailed explanation of the experimental setup should be provided to improve the clarity, and to ease other researchers when reproducing your result. 2) It might be clearer to discuss the HF and FO variants of Per-FedAvg in the methodology part. 3) What is the CNN model for MNIST and CIFAR-10 tasks?

Relation to Prior Work: Yes.

Reproducibility: No

Additional Feedback: 1) In L120: typo "was has been" 2) Why using D^i_t, D^{\prime i}_t, D^{\prime\prime i}_t in Eq. 8? What will happen if we use the same batch of data computing the three parts of Eq. 8? 3) What is the performance of FedAvg without "update"? 4) Can you provide training time statistics for different methods?

[Author Response · NeurIPS 2020]



(a) Fig. 1

(b) Fig. 2

We thank the reviewers for their careful consideration and constructive feedback. Below, please find our responses.

**Reviewer 1.** **Possibility of using a decreasing stepsize:** Indeed, it is possible to achieve the same complexity bound
using a diminishing stepsize. In particular, by using $\beta_k$ as the stepsize at iteration $k$, eq. (109) holds with $\beta = \beta_k$.
Hence, summing up this equation for $k = 0, ..., K-1$, we recover the same complexity bounds using $\beta_k = \mathcal{O}(1/\sqrt{\tau k})$.
We'll mention this point as a remark in the revised paper. **Concerning the term $\alpha^2 \sigma_G^2/D$ & comparison with [37]:**
Note that the term $\alpha^2 L^2 \sigma_G^2/D$ appears in the upper bound due to the fact that $\tilde{\nabla} F_i(w)$ is a *biased* estimator of $\nabla F_i(w)$.
In particular, as shown in Lemma 4.3, the bias is bounded by $\alpha L \sigma_G/\sqrt{D}$. This bias term will be eliminated if we
assume that we have access to the exact gradients at training time (see the discussion after Lemma 4.3), which is the
case in [37] where the authors focus on the deterministic case. We'll make the differences with [37] clearer in the
revised version. Thank you for your suggestion. **Questions regarding the numerical experiments:** Please see Fig. 1
which illustrates the average test accuracy of all studied algorithms with respect to time. We will include this in the final
version of the paper. **Regarding lines 266-268:** We will clarify the data distribution using a figure.

**Reviewer 2.** **Novelty of the paper and comparison with other theoretical results:** We'd like to emphasize that the
main contribution of our work is to provide the first convergence guarantees for meta-federated learning algorithms in
the model-agnostic meta-learning regime and for non-convex functions. In particular, [32], mentioned by the reviewer,
focuses on the analysis of MAML for centralized settings, and hence, it does not include the local updates on each
node ($\tau > 1$) which is one of the main challenges in the analysis of FL algorithms in general. In fact, Proposition
F.1 is stated to deal with this challenge which does not exist in the centralized setting at all. In addition, we'd like to
add that our analysis is totally different from other meta-federated learning works, such as [38], since they consider
different meta-learning regimes. Moreover, it is worth noting that [38] focuses on strongly-convex functions while we
study non-convex objective functions. **Regarding comparison with other algorithms such as [38]:** Following your
suggestion, we also compare our method with ARUBA. To do so, we also report the output of FedAvg+ARUBA after
refinement for each user. In particular, we consider $\tau = 4$ and $K = 1000$, and also tune hyper-parameters of ARUBA for
a fair comparison. The final accuracy of all algorithms is as follows: Per-FedAvg(FO): $34.1 \pm 0.08$, Fed-Avg+ARUBA
(with refinement): $36.74 \pm 0.1$, Per-FedAvg(HF): $43.71 \pm 0.12$. In Fig. 2, we have also depicted *one realization* of
training path, just to provide intuition on the convergence speed of these methods. **Regrading $\epsilon$-stationary definition:**
The reviewer is right that there is an inconsistency here. We'll update our definition as $E(\|\nabla f(w)\|^2) \le \epsilon$ to make it
consistent with the result of Corollary 4.6. Thanks for catching this typo. **Dependence of Wasserstein distance on**
**dimension:** The reviewer is right that the convergence speed of Wasserstein distance in d dimension is exponentially
slow. Our main goal was to elaborate on the dependence of Wasserstein distance on the number of samples. We will
clarify this matter. In addition, we'd like to highlight that our result on TV distance does not suffer from the same issue.
Thanks for raising this point.

**Reviewer 3.** **NLP Example:** NLP example is mainly mentioned in the introduction to highlight the role of data
heterogeneity. Indeed the same story holds for images stored on users' devices, which is more consistent with
our experiment. We thank the reviewer for this suggestion, and to complete the story, we will add a language
model experiment as well. **Regarding the clarity of the paper:** We will provide more details about the setup of
our experiments and also clarify the points that the reviewers brought to our attention. **Regarding details of the**
**experiment:** The reviewer is right about the experiment setup (distribution of images). We will clarify our setting, and
will also add a figure to explain the distribution of images better. We have also provided the code and will include the
updated code in the final version as well. We have used a fully connected neural network with two hidden layers in this
experiment. Thanks for your feedback. $D_t^i, D_t'^i, D_t''^i$ **in Eq. 8:** For the sake of analysis, we need these datasets to be
independent. That's why we use different datasets. We'll highlight this point. **Experiments:** Please see Figure 1 for the
performance of Fed-Avg with and without update, Per-FedAvg (FO), and Per-FedAvg (HF) with respect to time.

[Meta-Review · NeurIPS 2020]

The reviewers agree that the paper is solid and make novel contributions to understanding federated learning and meta-learning.